# Data-Driven Controller for Drivers' Steering-Wheel Operating Behaviour in Haptic Assistive Driving System

Simplice Igor Noubissie Tientcheu [1], Shengzhi Du [1,*], Karim Djouani [1] and Qingxue Liu [2]

[1] Department of Electrical Engineering, Tshwane University of Technology, Pretoria 0183, South Africa; 210179810@tut4life.ac.za (S.I.N.T.); djouanik@tut.ac.za (K.D.)
[2] School of Mechanical and Electrical Engineering, Kunming University, Kunming 650208, China; qxliu@kmu.edu.cn
[*] Correspondence: dus@tut.ac.za

**Abstract:** An advanced driver-assistance system (ADAS) is critical to driver–vehicle-interaction systems. Driving behaviour modelling and control significantly improves the global performance of ADASs. A haptic assistive system assists the driver by providing a specific torque on the steering wheel according to the driving–vehicle–road profile to improve the steering control. However, the main problem is designing a compensator dealing with the high-level uncertainties in different driving scenarios with haptic driver assistance, where different personalities and diverse perceptions of drivers are considered. These differences can lead to poor driving performance if not properly accounted for. This paper focuses on designing a data-driven model-free compensator considering various driving behaviours with a haptic feedback system. A backpropagation neural network (BPNN) models driving behaviour based on real driving data (speed, acceleration, vehicle orientation, and current steering angle). Then, the genetic algorithm (GA) optimises the integral time absolute error (ITEA) function to produce the best multiple PID compensation parameters for various driving behaviours (such as speeding/braking, lane-keeping and turning), which are then utilised by the fuzzy logic to provide different driving commands. An experiment was conducted with five participants in a driving simulator. During the second experiment, seven participants drove in the simulator to evaluate the robustness of the proposed combined GA proportional-integral-derivative (PID) offline, and the fuzzy-PID controller applied online. The third experiment was conducted to validate the proposed data-driven controller. The experiment and simulation results evaluated the ITAE of the lateral displacement and yaw angle during various driving behaviours. The results validated the proposed method by significantly enhancing the driving performance.

**Keywords:** haptic guidance; backpropagation neural network; genetic algorithm PID; driving behaviour; fuzzy-PID controller





## 1. Introduction

Although driving is essential to everyday life, road accidents are one of the most common causes of fatality and economic loss in society [1], causing the loss of life of more than 1.3 million and 50 million injuries yearly [2]. Road conditions such as weather and vehicles with human interaction are external factors contributing to road accidents [3–6]. Some studies have discovered that driver errors and characteristics are the leading cause of fatal vehicle crashes [7–9]. According to the European Union, the deaths engendered by car accidents result from vehicles being out of their lane [10]. To mitigate a significant number of road accidents, many researchers have developed new assistive technology, from driver assistance systems (DASs) to advanced driver assistance systems (ADASs), with the purpose of improving driving behaviour errors and performances. A lane-keeping assistance system (LKAS) and advanced lane-keeping assistance (ALKAS) have been developed to assist drivers in staying in their lanes by improving their position [11,12].

These techniques were used to prevent head-on crashes and 27% of fatal accidents were avoided due to the implementation of this technique [13,14].

Studies were conducted on driver assistance systems. Continuous haptic steering guidance was developed to constantly assist the driver steering wheel with force feedback in response to centre lane deviation or by alerting the driver with vibrotactile devices mounted on the seatbelt and steering wheel once the vehicle changes its lane or goes out of the predefined lane with the aim of improving the driver reaction and speed [15–18]. These works were validated as effective in lowering the driver's workload and improving the target trajectory achievement. However, this haptic guidance for lane-keeping assistance was developed based on the car–road model but did not consider human behaviour. The human uncertainty could limit performance. The driving behaviours need to be understood and predicted to achieve an excellent assistive system.

Several researchers have modelled human driving behaviour in the car–road closed-loop haptic system by predicting the driver intention with the aim of enhancing driving performance during lane-keeping and lane-changing tasks [18–21]. The link between the driver's vision and steering behaviour with force feedback was investigated by Franck Mars [22] when driving through the curve. Nevertheless, their output models were based on driving intention variables. The driving behaviour and the vehicle dynamics were modelled from mathematical principles, with many assumptions leading to instability. Different controllers were developed to adjust the uncertainty or reduce the lateral displacement by compensating for the driving behaviour and yaw angle error.

Model predictive control (MPC) is a feedback control algorithm that uses the model to optimise and predict the future output of the system. A multivariable controller monitors the work simultaneously by considering all interactions between system variables. It can consider constraints, such as speed limit, lateral range on the input and more system states. Based on the above characteristics, researchers have used MPC to compensate for driving behaviour by providing a steering action to minimise the cost function involving car path errors and dynamics. Shivaram [23,24] proposes a two-MPC approach to the keeping feature to reduce the vehicles' deviation from the centre lane. Bujarbaruah et al. [25] developed an adaptive MPC for lane keeping with many constraints on the steering wheel angle and the roadway limit to minimise the lateral displacement error. To minimise the steering angle error, Keen and Cole [26] designed a steering compensator derived from a linear MPC by applying the formal system identification technique to the double lane-changing tasks. Pick et al. proposed an MPC combined with PID to control the driver's steering, braking and accelerating by optimising a cost function that considers the lateral errors and speed deviations [27]. However, despite the aforementioned studies addressing the issue of constraints, the vehicle's mathematical assumptions and the adaptation of various driver uncertainties in real time become problematic. Moreover, the driving behaviour was modelled as a linear and time-invariant system. The driver–car–road system is complex in a real environment, with various time-varying driver tasks and actions. Therefore, these control approaches should consider the system non-linear and time-varying.

To predict the driving behaviour control action and minimise the lateral displacement error, Guo et al. [28] combined MPC and PID [29]. The PID controller was designed to compensate for the driver's braking and accelerating control by minimising the longitudinal velocity error. The neural response system and neuromuscular characteristics of the driver were taken into consideration. The MPC developed in [30] was further enhanced by including the vehicle inverse information in the longitudinal driver control. This approach was expanded using the MPC to control a non-linear time-variant driver steering operation. The driver steering controller was developed and the driving steering behaviour model was used as a non-linear time-varying controller. A non-linear vehicle dynamic with the path preview and state feedback was incorporated to solve an MPC loss function and produce a driving command. However, the non-linear dynamics of the vehicle were approximated by a set of linearised models and assumptions were made about different driving skills (highly skilled, less skilled and novice driver steering).

Stochastic MPC (SMPC) designed for a driver steering controller which considered the road condition (icy, wet and dry), preview point technique, preview time, weight condition, and time delay was proposed by Qu et al. [31]. The cost function, which involved the weighted combination of lateral path error and human driver control, was minimised by the SMPC. They used the disturbance of the internal vehicle with the dynamic road friction coefficient to design the driver steering knowledge on various road conditions. Although this control model has a good tracking path, it is essential to note that the road friction coefficient and vehicle internal dynamic formulations were based on mathematical and many other assumptions.

Some researchers also used Proportional-Integral-Derivative (PID) control to mimic and control the steering wheel behaviour. The Shida et al. studies developed a PID control approach with Particle Swarm Optimisation to improve the driver input torque in this system; the driving behaviour was modelled as a PID with conventional feedback, and the PSO minimised an evaluation function to find the best PID gain (Kp, Ki, Kd) that would reduce the driving model error and assist the driver to apply less effort in steering [32]. Nevertheless, the fixed control parameter was used, making the controller less robust. Menhour et al. [33] proposed two PIDs combined with a Linear Quadratic Regulator (LQR) to minimise the yaw angle and the lateral displacement error in the trajectory tracking task. An LQR and linear matrix inequality optimisation obtained the PID parameters. Due to its robustness and stability, the controller overcomes uncertainties. However, only the trial and error method was used; it relied on the accuracy of the driver–vehicle model and required a fast processor. Also, it could not handle various driving behaviours. A PID controller was also designed by Niu [34] and Sun to monitor the robot driver's accelerator leg. The parameters were derived from a critical proportioning method obtained from the mathematical transfer function of a robot's accelerator leg. The controller cannot resist uncertainty and a non-linear system because the driving behaviour model was made from a mathematics approach with many assumptions. A fuzzy-PID controller was proposed to reduce the displacement error and match the tracking error. The fuzzy controller was used to improve the overshoot (decrease) and the response speed (reduce the rising time) [35]. But it needed a vast, distinct rule base. The fuzzy scale factor is difficult to adjust.

Ercan et al. [36] designed an MPC to support the driver in a haptic-shared control system to keep the vehicle in the lane and reduce the workload. The driver arm impedance model has been considered for the loss function. The controller estimated and updated the best guidance torque, which assisted the driver in keeping in the lane by reducing their workload. An advanced MPC-based haptic controller combined with LQR was developed to enhance driving behaviour based on cognitive processes. The future state of the driving model was forecast by a cognitive controller involving an LQR. The cost function with various constraints on steering angle, yaw rate, lateral velocity and torque assistance was optimised to provide an assistive torque during a steering action [37]. Efremov et al. [38] developed an MPC controller that used driver environment restriction to support the driver in lateral tasks. The main aim of the MPC controller was to predict the optimised guiding steering wheel angle from a reference steering angle conducted by the driving action. In a curvature, the sideslip angle and the slip ratio on the tyre forces were delimited and taken as a constraint for the MPC objective function. For the longitudinal control, Yang et al. [39] developed an MPC to alternate human drivers and the assistive driving system to reduce the driver workload. This system has considered as a constraint the switching frequency between the system and the driver and, additionally, the human factor out of the loop was considered when driving with the assistive system only. However, the driver arm impedance model was derived from mathematical assumptions; therefore, it will lack robustness. Furthermore, some controllers did not consider an optimal human model, which could not compensate for different driving steering behaviour errors or styles.

It is also crucial to mention that vehicle rollover and sideslip impact driving behaviour and, therefore, car instability, contributing to a significant number of fatal accidents [40,41]. Fernando et al. [42] developed an IoT (Internet of Things) system to estimate the precise

vehicle rollover and sideslip angle by taking into account the communication delay, with the aim of improving driving comfort and safety. Arslan et al. [43] proposed a non-linear predictive controller that constantly diminishes the tracking error and avoids the vehicle rollover. However, their systems rely on the stability of the designed vehicle dynamics.

Although several controllers have been extensively evaluated to model and control driving behaviour in the above review, no study has investigated the data-driven controller considering various driving styles in haptic feedback systems. This study presents a proposed robust controller that uses the data-driven technique, the genetic algorithm offline, to minimise the fitness function and provide the best PID parameter. Moreover, this PID gains controller will be used as the output of the fuzzy-PID controller online due to the dynamic change in driving behaviour response to improve different driving performances by minimising the car's lateral displacement and orientation angle errors.

This paper is structured as follows: Section 2 presents the relevant works used to achieve the objective of this research work, such as the design of a driving behaviour model by an artificial neural network and the optimisation of the PID controller using GA. Section 3 formulates the problem, highlighting the proposed data-driven controller design and structure. Section 4 describes the experiment and driving simulator; the simulation and results are presented in Section 5. An extensive discussion is provided in Section 6. Lastly, the conclusion that synthesises and consolidates the state of the art is presented in Section 7.

## 2. Relevant Works

### 2.1. Artificial Neural Network Model for Driving Behaviours

Most driving behaviour models were designed on a model-based approach and the close-loop feedback signal was derived from the lateral motion mathematical model. These models' dynamic includes several parameters that are not elucidated, and the first principle of physics is applied, where many assumptions and simplifications arise. Driving behaviour is a complex task and should be considered as a non-linear and time-varying system. So, modelling human driving behaviour from mathematical and physical principles will decrease the matching performance between the model and the actual driving intention. An artificial neural network (ANN), viewed as an intelligent approach, can provide a model without prior knowledge of the driver–vehicle road system. Furthermore, it reacts and makes decisions in a similar way to the human brain [44]. Some studies have used the development of an ANN to build driving behaviour models to avoid vehicle collision [45], where the braking and steering data were deeply analysed. With the ability to learn and make decisions, many researchers implemented the ANN to match the driving behaviour in lane keeping and lane changing without any calculation [20,46–50]. The proposed model was accurate compared to the mathematical or feedback control model-based approach.

Backpropagation is an ANN training algorithm that minimises the deviation between the ANN output and the desired output by gradually adjusting the weights in all connections according to the desired output target in the node. Such a backpropagation-trained neural network (BPNN) is made of a multilayer feedforward neural network with three or more layers of interconnected neurons. It includes an input, hidden and output layer [51]. The framework of the BPNN is designed based on the characteristics of the input and output data. The hidden layer tends to be distributed between them, codifying each aspect of the input data. This paper will map the driving behaviour based on data from the driver and the vehicle provided.

Figure 1 represents the driving behaviour model with the haptic steering feedback force using the BPNN.

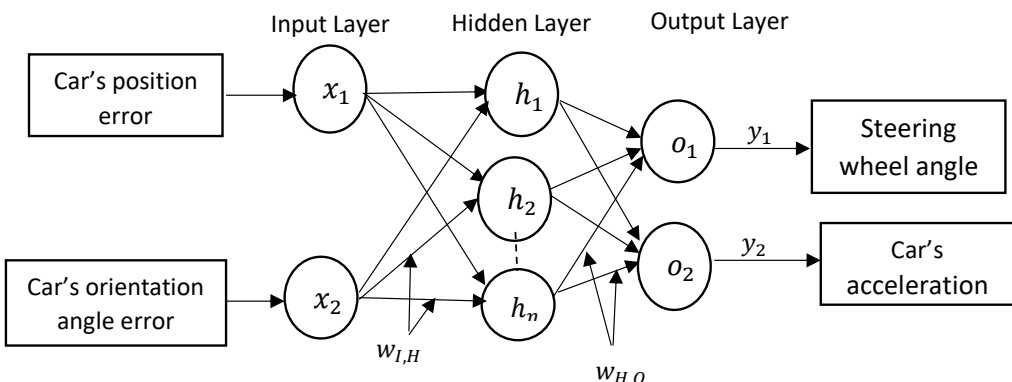

**Figure 1.** Driving behaviour model using BPNN.

This model is made with three layers, namely, two input units($x_1$, $x_2$), hidden units ($h_1..h_n$), and two output units ($O_1$, $O_2$), as indicated in Figure 1. The input variables of these BPNNs are the car's position error and orientation angle error. The steering wheel angle and acceleration are the output variables. The number of hidden neurons in the hidden layer is ten (10). The input neuron or units are joined to the hidden neuron through the weights $W_{I,H}$ and the hidden unit is connected to the output neuron through $W_{H,O}$. The input data of the BPNN is

$$X = [x_1, x_n] \tag{1}$$

and the BPNN output forward algorithm is defined by Equation (2)

$$y_j = f[\sum_i^N w_{ij}x_{ij}] \tag{2}$$

where $y_j$ is the actual output; $w_{ij}$ is the weight, and i represents the starting unit's (neuron) identifier and j the output unit's (neuron) identifier. The backpropagation algorithm is expressed in Equation (3) [52]. This is the adjusting calculation of the weights joined to the output.

$$\Delta w_{ij} = r \times \Delta o_j \times x_i \tag{3}$$

where $\Delta o_j$ is the error coefficient; and $r$ the coefficient; $x_i$ the actual input

$$\Delta o_j = (d_j - y_j) \times f'(y_j) \tag{4}$$

where $d_j$ represents the desired output and $y_j$ the actual output.

The achievement of the correction of the weights is formulated as follows

$$w_{ij(n+1)} = w_{ij(n)} + \Delta w_{ij} \tag{5}$$

### 2.2. Optimising PID Controllers Using GA

The PID controller is a popular controller used to enhance the transient response. In this paper, the PID controller utilises the feedback errors from the driver–vehicle system and constantly varies the steering wheel angle according to the car's position error and orientation angle error. The general PID controller used in this study is shown in Equation (8). The proportional control and integral control reduce the steady-state errors (deviation between the car's desired position and orientation and the actual position and orientation angle). The derivative controller improves the system response speed.

$$\Gamma(s) = (K_p + K_i \frac{1}{s} + K_d t)e(s) \tag{6}$$

where $e(s)$ is the difference between the car's actual lateral position and orientation and the car's desired position and orientation; $K_p$ represents the proportional gain, $K_i$ the integral gain and $K_d$ the derivative gain.

The genetic algorithm (GA) has been used in optimising PID controllers because it can explore a huge search region of PID gains and find the optimal set by optimising the system error or other performance indexes. The individual from a new generation reproduced by a population (collection of individuals) seems to produce a better performance than the previous individual [53].

The GA-PID was chosen due to the complexity of human driving behaviour and the non-linearity of the car–road–vehicle system with a haptic feedback interface. The GA combined with the PID compensator optimises the gains, minimising the BPNN driving behaviour error, including the car's position and orientation angle error. One set of PID gains represents one of the individuals in a GA population. The optimisation technique assesses each parameter set due to a loss function, integral time absolute error (ITAE), on the car's lateral displacement and orientation error, indicating the PID controller's performance. A new generation generates a recent PID gain set using the GA's selection, cross-over and mutation action. The action will repeat until the optimal PID gains set minimises the loss function (ITAE).

## 3. Proposed Data-Driven Controller

This controller is designed based on the drivers' data profiles because of the non-linearity and the complexity of the individual human behaviour and personality. The BPNN representing the driver was mapped using real data collected from various driving behaviours or scenarios. The GA used this driving behaviour model to optimise the ITAE cost function to provide the optimal PID parameters offline, as shown in Figure 2. This figure depicts the diagram of the GA-PID structure used in this paper, where the GA is used to produce the best PID parameters ($K_p$, $K_i$, $K_d$) that provide the best driving steering angle $\alpha_{sw}$ by compensating for the driving error ($e_p$, $e_o$) committed by the driver whose model was mapped by a BPNN. This controller minimises the cost function, i.e., ITAE, as shown in Equation (10). The optimal PID parameters emanating from different driving behaviours were then used to map the inputs and the outputs of the membership function of the fuzzy-PID controller integrated into the system to operate online.

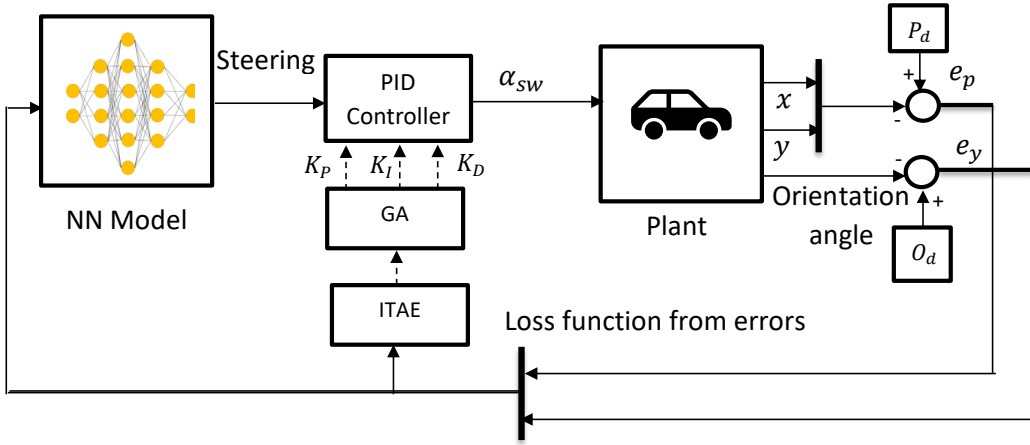

**Figure 2.** Driver behaviour controller using GA-PID.

### 3.1. Modelling Driving Behaviour Using BPNN

As mentioned before, due to the complexity and the non-linearity of driving behaviour, the BPNN training algorithm was chosen to map these behaviours because it consistently adjusts the weights in all connections according to the desired output target to minimise the error between the driver's ANN output and the desired output. The driving behaviour designed model is presented in Figure 1 where the data emanated from driving tasks.

### 3.1.1. Data

The real-time data were collected from MATLAB Simulink after conducting multiple driving simulator experiments with different drivers. These data were classified based on their performances.

- Input and output data

The driving behaviour is characterised by the driver's movement on the steering wheel, the acceleration, the car's lateral position, and the vehicle orientation angle error. The BPNN will be framed based on the input variable, such as the car lateral deviation, the car orientation angle error and the speed. The output variables are the steering wheel angle and the acceleration.

- Training, validating and testing data

Training data is a portion of real data used to train the BPNN model. Validating data are samples of original data used to evaluate the BPNN model while testing data are used to examine the correctness of the model. In this work, 75% of the real data generated by the drivers and the vehicle (car position, speed, acceleration, orientation angle and steering wheel angle) are randomly selected for training data, 15% of data are used for validation and 15% for testing.

### 3.1.2. Data Processing: Normalised and Denormalised Data

Normalisation in BPNN can be seen as the data preparation procedure. It primarily converts a given data set's input or output value to a scale without affecting input variation [54]. Z-score normalisation, a normalisation technique, has been used in this paper.

- Z-score normalisation technique

This method provides a Z-score value that shows how far a data point is from the mean value, as shown in Equation (6).

$$Z = \frac{x - \mu}{\sigma} \tag{7}$$

where $x$ is the evaluated value, $\mu$ is the mean and $\sigma$ is the standard deviation. According to Mohammed Z, Z-score standardisation in BPNN speeds up the learning process by reducing the number of epochs [55]. This paper applied Z-score normalisation on the input and output data since the range in the input (displacement and orientation errors) was significant. The range of target data (steering wheel angle and car acceleration) was substantial.

- Denormalisation technique

Denormalisation data is a technique that is used to retrieve the original data which was subjected to normalisation. This approach links the BPNN to the physical. Equation (7) is used to recover the original data structure, such as the steering wheel angle and the car acceleration data.

$$x = (z * \sigma + \mu) \tag{8}$$

where $x$ is the retrieved value, $\mu$ is the mean and $\sigma$ is the standard deviation.

### 3.2. GA-PID Structure for Controller Design

In a PID controller, the parameters are the key to improving the system response, which in this study are optimised by GA in this paper.

#### Genetic Algorithm

The GA algorithm is a mathematical approach to solve sophisticated or constrained optimisation matter based on biological evolution [56]. In the genetic algorithm process, an offspring is generated for the next generation from parents chosen randomly from a population [57,58].

- Fitness Function

  In a genetic algorithm, the fitness function is an index to assess the chromosome that can produce one or more children for the generation [59]. In this work, the objective function of GA is defined based on the performance index ITAE. Optimising this function provides the optimal PID parameters. The ITAE shown in Equation (9) is the criterion used to evaluate the performance of a PID controller. In this minimisation problem, the fittest PID parameter will indicate the lowest numerical value (ITAE) related to the cost function J [53,60].

$$ITAE = \int_0^T t(|e(t)|)dt \tag{9}$$

In this paper, the ITAE of the car's lateral position and the car's orientation angle are used as the cost or objective function, as in Equation (10)

$$J = \int_0^T t(|e_p(t)| + |e_{or}(t)|)dt \tag{10}$$

where $e_p$ is the difference between the desired car's position and the actual position, $e_o$ constitutes the deviation between the target car's orientation angle and the actual orientation angle and $T$ illustrate the optimisation simulation time. The loss function J will be minimised with the following constraints for each driving behaviour:

$$\begin{cases} K_{p_{min}} \leq K_p \leq K_{p_{max}} \\ K_{i_{min}} \leq K_i \leq K_{i_{max}} \\ K_{d_{min}} \leq K_d \leq K_{d_{max}} \end{cases} \tag{11}$$

The GA applied to this controller comprises three genetic operators: selection, cross-over, and mutation.

- Selection

  Selection in GA is the technique of choosing individuals from the population (PID parameters). The selection was based on fitness values derived from the fitness function J (Equation (10)). The probability of selection is higher for the fitter solution because it will generate more offspring. This selection can also be made randomly.

- Cross-over

  After the selection process, the cross-over algorithm exchanges some part of the selected strings and the genetic procedure manipulates the property of a chromosome to obtain the better part of the earlier (old) chromosome to produce the best new generation. Cross-over combines two parents' genetic information (chromosomes) to produce a new generation (offspring or new PID parameters), which is finding the best solution. The newly generated chromosomes are supposed to be better than their parent. During the cross-over technique, 100% probability indicates that a new generation will be generated from the selected generation and 0% means that the latest generation will be the same as their parents.

- Mutation

  After the selection and cross-over process, the wrong choice of the selected population and the lack of diversity in the initial strings can limit the search of GA from an ample space. The mutation operation can prevent the above-enumerated problem by changing a bit in a genetic sequence from its initial position. It prevents a population of genetic information (chromosomes) from being similar. This process is used to extend the GA searching space.

  Figure 3 is the flowchart which illustrates the GA PID controller. The population (PID parameters) is first initialised in the GA-PID structure. Then, the GA is applied through selection, cross-over and mutation to explore the best gains of the PID controller. If the best criteria are not satisfied, the new generation will replace the old one and the process will repeat until the stopping criteria are met.

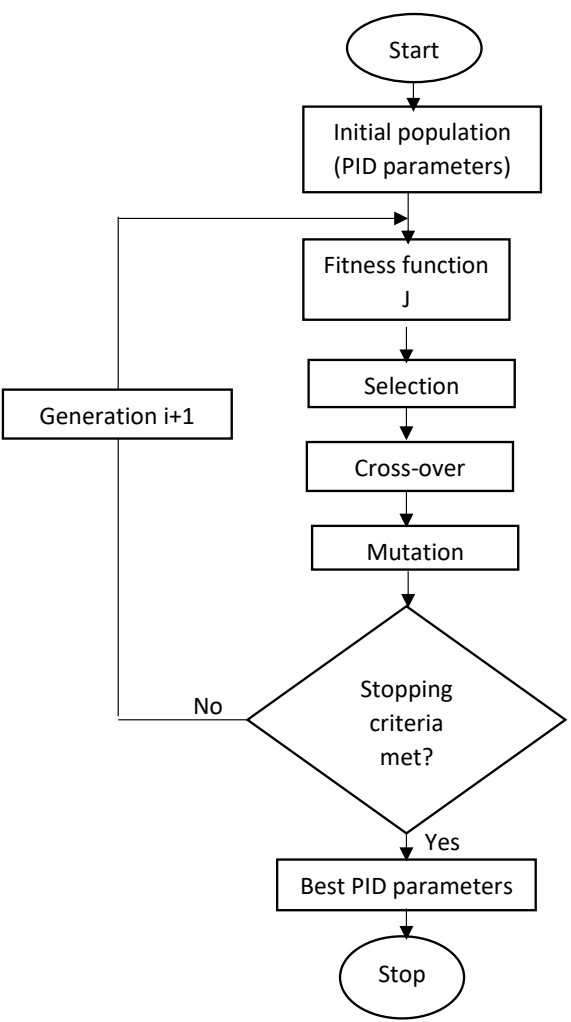

**Figure 3.** Genetic algorithm flowchart of GA-PID.

### 3.3. Fuzzy-PID Controller

The GA-PID applied to the driver steering wheel angle enhanced the performance by reducing the car position and car orientation angle error in the lane-keeping task. However, the personality variation in different driving behaviours and the dynamic change in operating conditions (speed, position error and orientation error) make this controller less robust. For each scenario or driving behaviour, the GA optimises the lateral position and orientation angle error based on BPNN behaviour by providing the optimal PID parameter sets. We proposed a fuzzy-PID controller where the fuzzy logic's output membership function is defined based on optimal PID parameters obtained by the genetic algorithm discussed earlier for different driving scenarios. This controller integrates the fuzzy logic technique into the PID controller. The fuzzy logic part of the controller automatically tuned the PID parameters ($K_p$, $K_i$, $K_d$) for different driving behaviours and various driving profiles, such as the car's position error, the car's orientation angle error and the car speed with haptic feedback forces to improve its robustness (flexibility and compatibility). This fuzzy-PID controller aims to account for the imprecision and uncertainties in different driving behaviours and profiles. Figure 4 is the block diagram illustrating the proposed fuzzy-PID compensator. The steering wheel angle ($\alpha_{sw}$) is directly affected when the PID gains are consistently adjusted using the online fuzzy logic rules; therefore, it will impact the car's lateral displacement and orientation angle. The human driver in this diagram replaced the BPNN that was trained using the data of the same driver in Section 2.

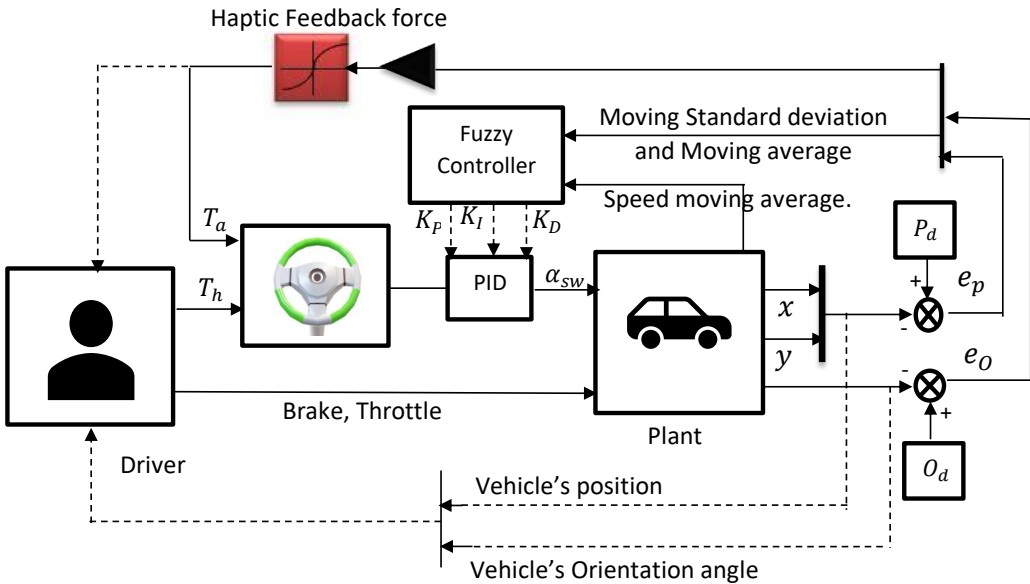

**Figure 4.** Various driving behaviour controls due to fuzzy-PID controller.

The fundamental structure of the fuzzy logic controller is shown in Figure 5.

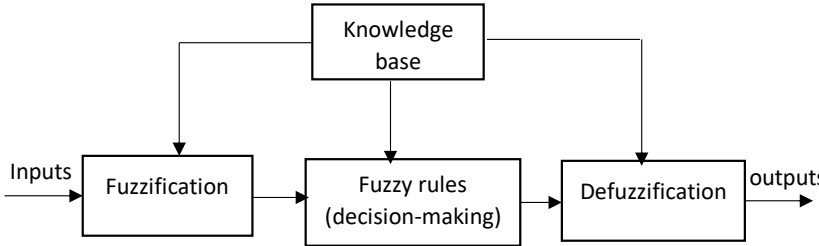

**Figure 5.** Basic structure of fuzzy logic.

The knowledge base was acquired after the GA optimised the parameters of the PID parameter sets.

### 3.3.1. Fuzzification

Fuzzification is the action of converting numerical input data into linguistic terms [61]. The car's lateral position error, orientation angle error and speed are driving profiles used as the input to the fuzzy controller, representing the membership function. Each membership is a fuzzy set divided between three (3) and four (4) fuzzy subsets given in the linguistic terms ($L$, $M$, $H$, $VH$). The fuzzy subsets of this controller were derived from various driving behaviours or scenarios. The fuzzy subsets were selected based on how (low, medium, high or very high) the car's lateral position error, orientation angle error and speed affected the driver's performance. The large base of some fuzzy subsets is the vast data's representativeness capacity, while the narrow ones represent sensitive data. In this paper, the triangular membership function has been chosen and input membership is illustrated in Figure 6.

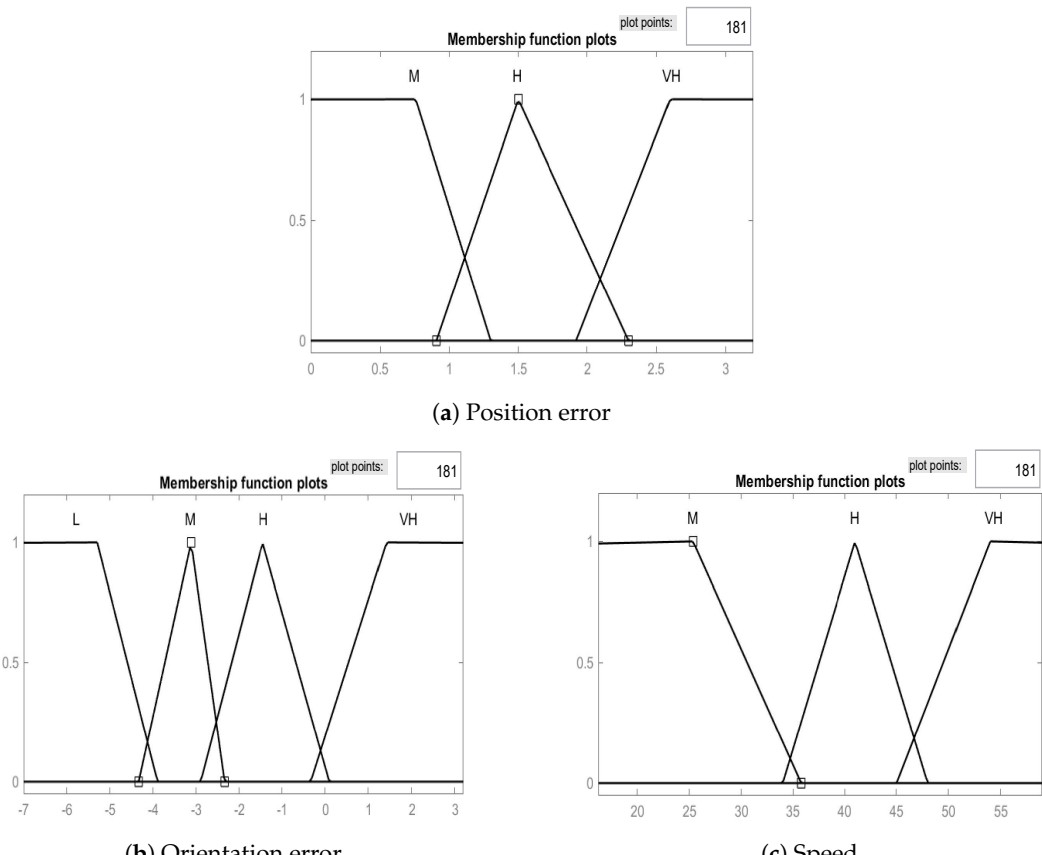

(**a**) Position error

(**b**) Orientation error             (**c**) Speed

**Figure 6.** The membership function of car displacement (**a**), orientation error (**b**) and speed (**c**).

The output variables of the fuzzy-PID are the PID parameters. The output membership function of the fuzzy controller is made of the PID parameters ($K_p$, $K_i$, $K_d$), and these parameters were automatically calculated according to the driving scenarios or driving behaviours optimised by the GA. Each parameter is defined by a fuzzy set divided into five (5) fuzzy subsets with different linguistic terms ($K_{VL}$, $K_L$, $K_M$, $K_H$ and $K_{VH}$) as indicated in Figure 7. The output and the input of the system have a linguistic relationship. The variation in these PID parameters from one driving style to another is better for providing compatibility and flexibility for the controller.

### 3.3.2. Fuzzy Rule Base

The fuzzy rule base in the fuzzy controller is used to model the interaction between the input and output fuzzy set by using an IF–THEN structure. The output fuzzy variables (PID parameters $K_p$, $K_i$, $K_d$) are determined by the input membership function ($e_p$, $e_o$, speed) by the fuzzy rule IF–THEN. A fuzzy rule base is given in the following form.

$$R_k: \begin{cases} \textbf{IF } e_p \in A_{k1} \textbf{ and } e_O \in B_{k1} \textbf{ and } speed \in C_{k1} \\ \textbf{THEN } K_{pk} \textbf{ is } D_{k1} \textbf{ and } K_{ik} \textbf{ is } E_{ik} \textbf{ and } K_{dk} \textbf{ is } F_{dk} \end{cases} \tag{12}$$

where $R_k$ is the $k$-th fuzzy rule; $A_{k1}, B_{k1}, C_{k1}$ are the fuzzy subset inputs of the car's lateral position error, the car's orientation angle error and the fuzzy subset of the vehicle's speed, respectively; and $D_{k1}, E_{ik}, F_{dk}$ represent the fuzzy output control variables of the proportional gain ($K_p$), integral ($K_i$) and derivative gains ($K_d$).

In this process, the Mamdani [62] fuzzy inference system is used to map input membership to output because it can interpret the linguistic terms used, is easily integrated, and can control uncertainties and non-linear systems. The fuzzy logic uses the linguistic representation of the input fuzzy subset and integrates the fuzzy rules to produce the output linguistic variable based on the respective fuzzy subset values as presented in Table 1. Each

driving behaviour has its fuzzy rule based on the changes in car position error, orientation angle error and speed. This rule base is used to produce the best driving response.

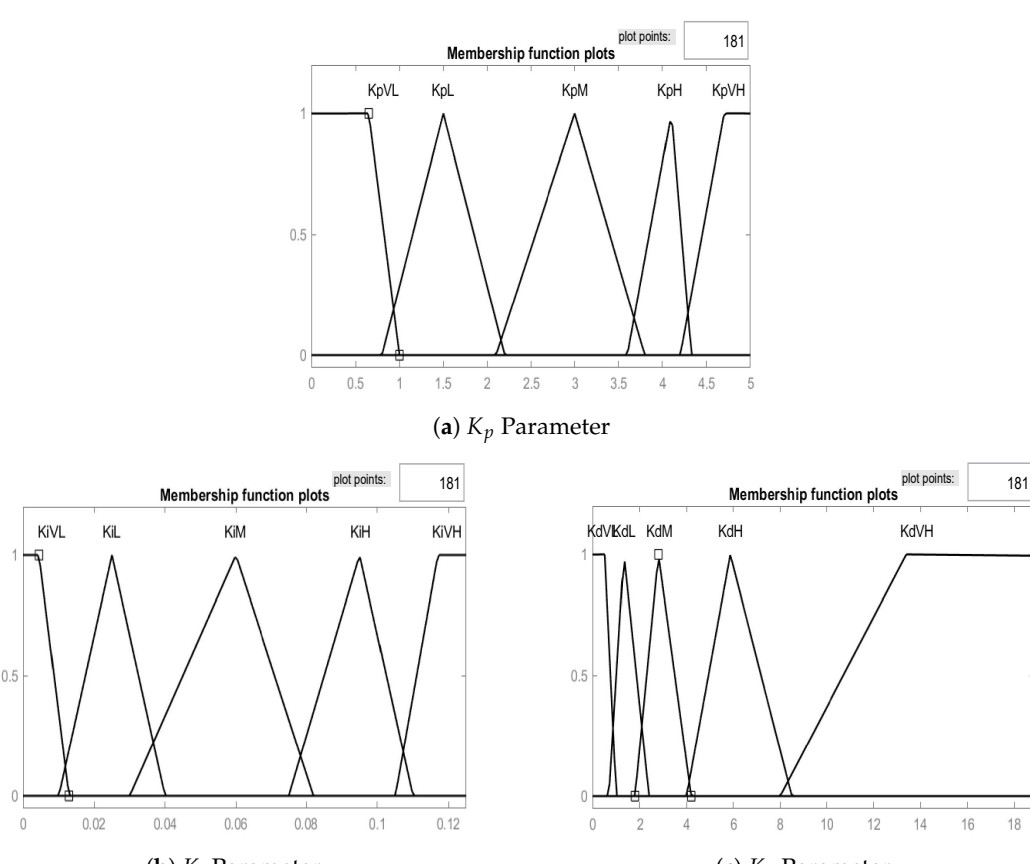

**Figure 7.** The membership function of $K_p$ (**a**). The membership function of $K_i$ (**b**). The membership function of $K_d$ (**c**).

**Table 1.** Fuzzy logic control rule base sample.

| Input Variable | | | Output Variable | | |
|---|---|---|---|---|---|
| **ep** | **eo** | **Speed** | **Kp** | **Ki** | **Kd** |
| L | M | H | KpH | KiL | KdL |
| L | H | H | KpH | KiL | KdL |
| L | VH | H | KpVH | KiM | KdVL |
| M | M | H | VL | VL | H |
| M | VH | H | VL | VL | H |
| H | M | M | VH | M | VL |
| L | M | L | M | H | VH |
| H | M | L | VH | VL | M |
| H | H | L | VH | VL | M |
| L | l | H | M | H | VH |

### 3.3.3. Defuzzification

Defuzzification is the opposite technique to fuzzification, which is the process of obtaining a converted exact value from the fuzzy set derived by inference and considered as the output [63]. In this study, defuzzification will map the fuzzy subset ($D_{k1}$, $E_{ik}$, $F_{dk}$) of the system and the appropriate membership degree to provide the PID parameter results in a single value. Based on the study developed in a fuzzy control system by Chen et al. [64], the equation illustrating the mathematical formula that produces the final output value by

defuzzification was derived in this study. The method used to compute the fuzzy values of the output is indicated in Equation (13).

$$K_{(p,i,d)} = \frac{\sum\limits_{i=1}^{N} \mu_i \cdot \beta_i}{\sum\limits_{i=1}^{N} \mu_i} \qquad (13)$$

where

$$\mu_i = min(A_{k1}(MSD_{e_p}), B_{k1}(MA_{e_p}), C_{k1}(MA_V)) \qquad (14)$$

$A_{k1}(MSD_{e_p}), B_{k1}(MA_{e_p}), C_{k1}(MA_V)$ are, respectively, the membership values of the subsets $A_{k1}$, $B_{k1}$ and $C_{k1}$; $\beta_i$ represents the single variables $(K_p, K_i, K_d)$ of the fuzzy output of $D_{k1}, E_{ik}, F_{dk}$.

### 4. Experiment Set up and Driving Simulator

This research work used a driving simulator and three experiments took place. In the first experiment, five drivers, four males and one female aged between 23 and 34, participated in two scenarios. The data collected were used to map the driving behaviours, contributing to the design of optimal GA-PID parameters offline and, later, the design of the proposed fuzzy-PID controller. In the second experiment, seven (7) subjects participated in the same driving simulator aged between 19 and 34 with five males and two females. The last experiments involving the first five drivers were conducted to compare and validate the proposed data-driven controller and an existing one. All these drivers had driver's licenses and different years of experience. In the first scenario, the driver was asked to drive the vehicle by following a centre line (green line) as displayed in Figure 8 with a haptic feedback system and without any controller.

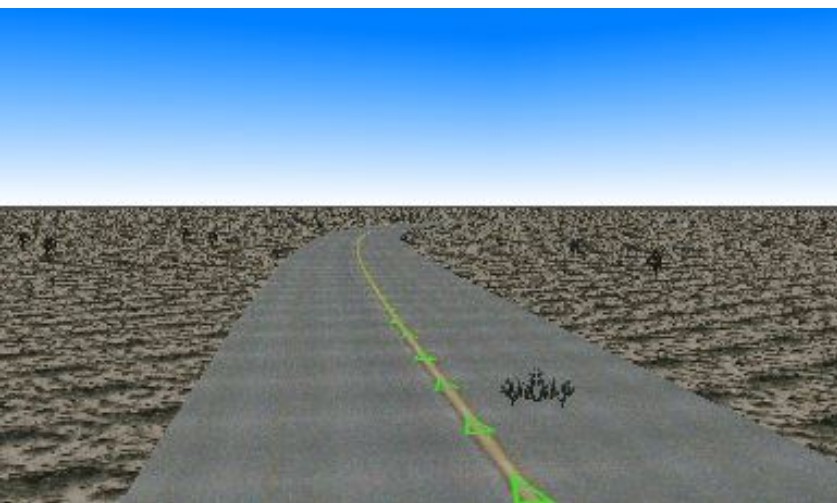

**Figure 8.** Line-keeping pathway.

After the first scenario, the same drivers were asked to drive the car by following a given centre line, the centre line with steering wheel haptic feedback and a data-driven controller. Figure 9 illustrates the participant driving in a complete simulator set.

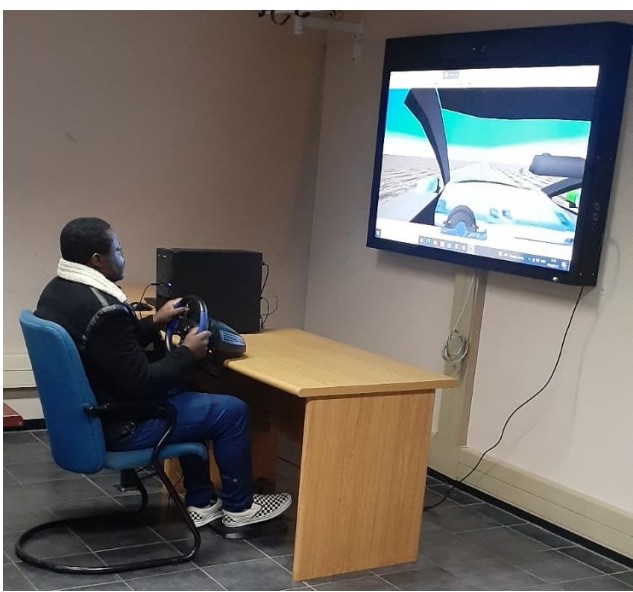

**Figure 9.** A subject participating in the simulator.

*4.1. Driving Simulator*

The stationary-based driving simulator was monitored by a computer and made of two key components, namely, a haptic feedback steering wheel (T150) force feedback mounted on a tabletop with two pedals (accelerator and brake) as shown in Figure 10.

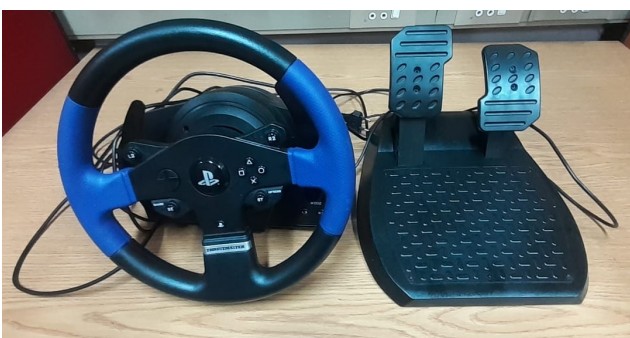

**Figure 10.** Haptic feedback steering wheel (T150).

The driving scene was displayed on an LCD monitor (resolution of 3840 × 2160) and the driver view option was selected as shown in Figure 11. The car driving simulator and environment used MATLAB Simulink, and the virtual reality on MATLAB Simulink ran on Windows 10.

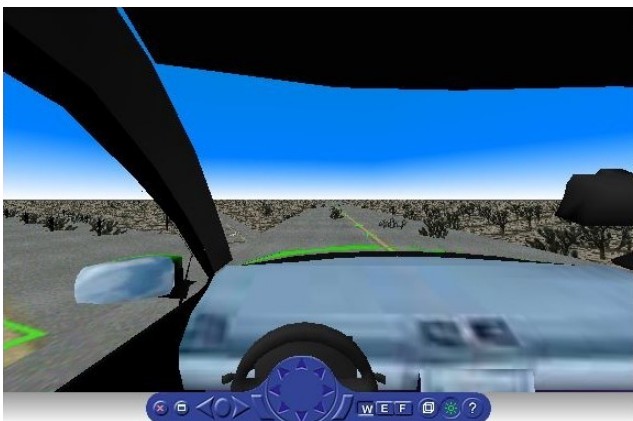

**Figure 11.** Stationary driving simulator: driver's view.

## 4.2. Computation Resources

The driving simulator was controlled by a haptic feedback steering wheel (T150) through an Intel core (TM) i5 10400 Dell computer (made in China) with a 64-bit operating system, a processor of 2.90 Hz and RAM of 8 GB. This gives the drivers a simple haptic steering wheel dynamic sensation. The simulation was performed in MATLAB Simulink 2020b, and the simulation time varied between 400 s and 500 s for different scenarios.

## 5. Simulation and Results

### 5.1. Driving Behaviour Using BPNN

After conducting the first experiment, the real data was collected through MATLAB Simulink. Different driving behaviours were mapped offline using the BPNN tool in MATLAB. The simulated results are presented in the following graph (Figure 12) and Table 2.

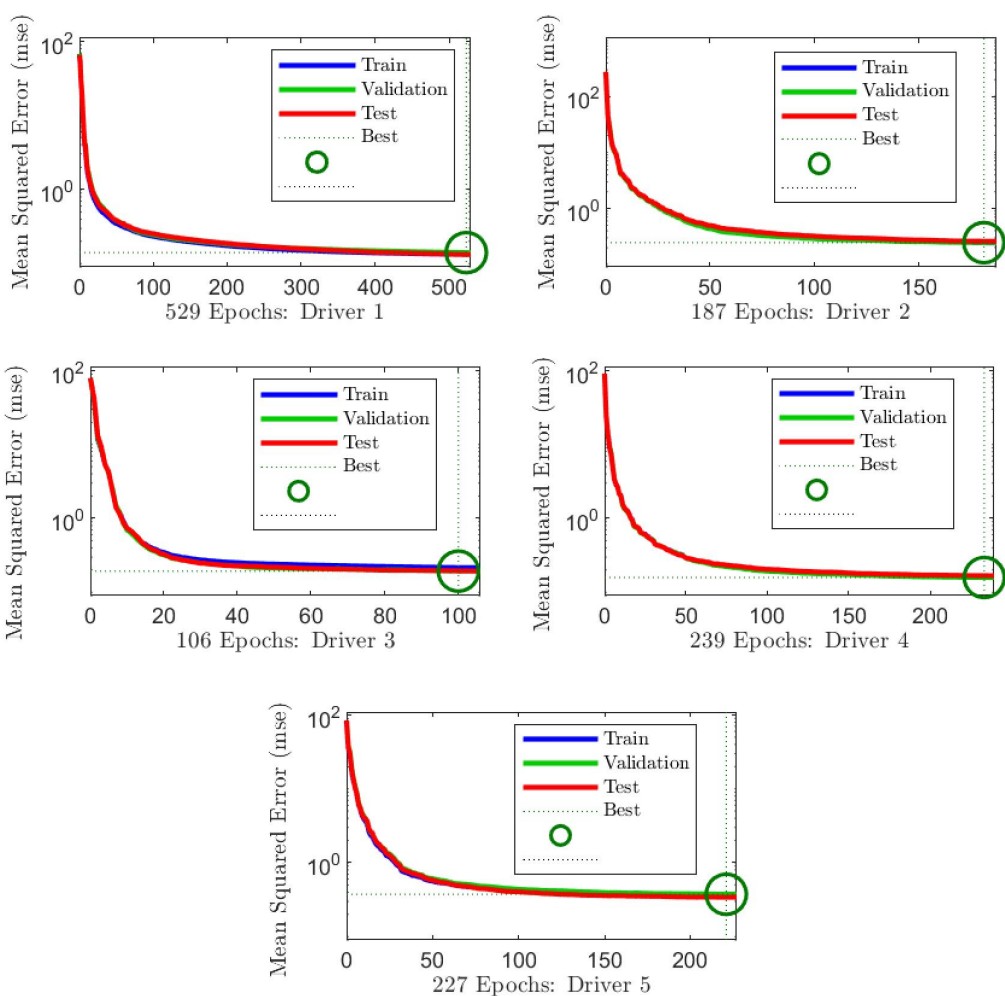

**Figure 12.** Performance plot of BPNN for different drivers.

Table 2 illustrates the performance analysis of BPNN for different driving behaviours as presented in Figure 12. Based on the Z-score normalised data method, the BPNN has a low MSE (mean squared error) for all the driving behaviours; these values vary and fluctuate between 0.14 and 0.37, indicating the accuracy of the BPNN. They all express an excellent regression.

**Table 2.** Performance analysis for BPNN with z-score normalised data.

| BPN Performances | Driver 1 | Driver 2 | Driver 3 | Driver 4 | Driver 5 |
|---|---|---|---|---|---|
| **MSE** | 0.14049 | 0.2452 | 0.19276 | 0.15823 | 0.37388 |
| **Epochs** | 523 | 181 | 100 | 233 | 221 |
| **Regression** | 0.93 | 0.869 | 0.886 | 0.916 | 0.803 |

5.1.1. Offline GA-PID Parameter Results

After mapping different driving behaviours, the GA-PID was simulated and the results providing the best PID parameters after optimising the ITAE loss function are shown in Table 3. Table 3 presents various optimised PID parameters for different driving behaviours. Figure 13 shows different BPNN driving behaviours due to car position with and without a GA-PID controller. The curve from driver 1 to driver 5 indicates that the driving behaviour without the GA-PID controller has a higher car position error with reference to the driving pathway. However, the system combined with GA-PID significantly improves car position and driver behaviour. The car position is closer to the desired pathway for all drivers, leading to less error.

**Table 3.** PID parameters obtained from a GA.

| PID Parameters | Driver 1 GA-PID | Driver 2 GA-PID | Driver 3 GA-PID | Driver 4 GA-PID | Driver 5 GA-PID |
|---|---|---|---|---|---|
| **Kp** | 3.3 | 1 | 4.904 | 1.313 | 4.987 |
| **Ki** | 0.108 | 0 | 0.08 | $3.81 \times 10^{-6}$ | 0.0035 |
| **Kd** | 13.543 | 6.866 | 2.399 | 6.891 | 4.416 |

Figure 14 represents various BPNN driving behaviour errors after a given task. This graph evidently shows that, for drivers 1 to 5, the error is lower for the BPN driving behaviour with the GA PID controller for the straight and corner roads as the road shape is indicated in Figure 13. This result also demonstrates the difference in the lateral trajectory of various BPNN driving behaviours indicated in Figure 13.

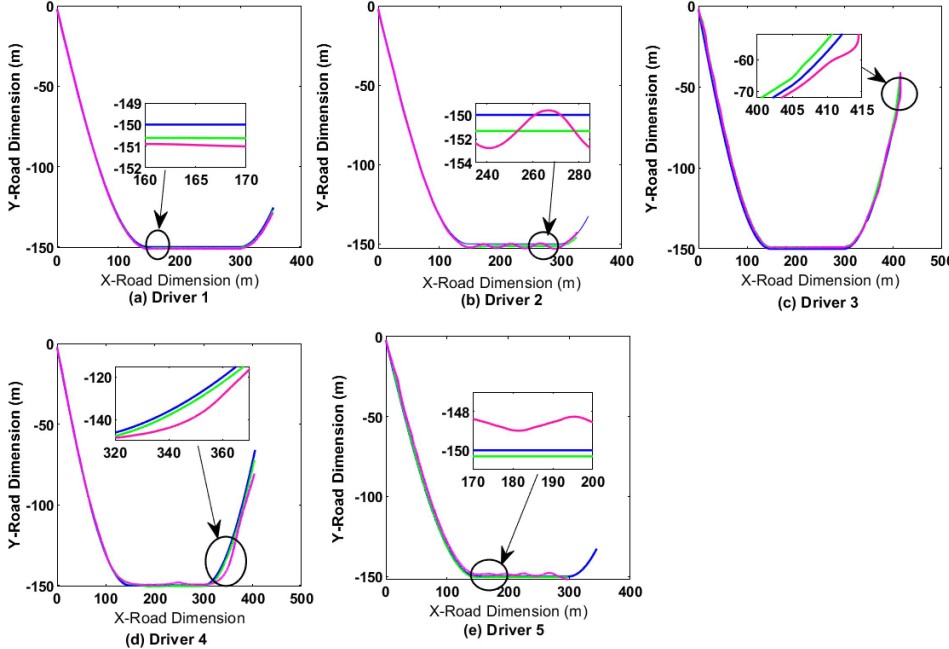

**Figure 13.** Various BPNN driving behaviour controls due to GA-PID controller after a given task (the blue (-), green (-) and magenta (-) colours represent, respectively, the desired pathway, the BPNN driving behaviour with GA-PID and the BPNN driving behaviour without GA-PID).

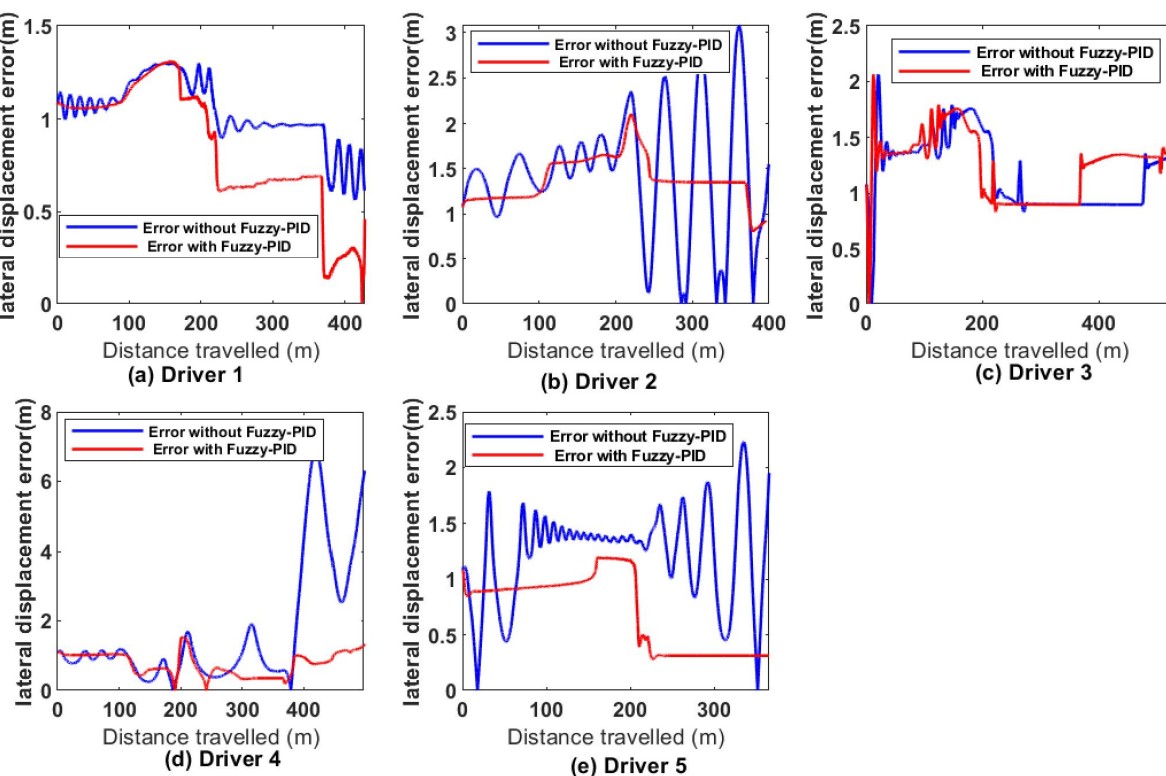

**Figure 14.** Car's position error using the BPNN model with and without the GA-PID.

5.1.2. ITAE for Car's Position Based on GA-PID Controller

The graphs displayed in Figure 15 result from the performance index ITAE (integral time absolute error) of different BPNN driving behaviours based on the vehicle car's lateral displacement after a given task. For this study, the lower the ITAE, the better the GA-PID performance is. The curve of the ITAE related to the BPNN driving behaviour combined with the GA-PID is significantly lower for drivers 1 to 5 compared to the system without the compensator. Table 4 illustrates the performance index (ITAE) on PID-GA on driving behaviour based on the car's lateral displacement, as mentioned above. The results enumerated in this table (Table 4) confirmed the results displayed in Figure 15. The car's lateral displacement error improvement in this table varies from 22% to 83%. After evaluating these results, we noticed that the PID-GA compensator notably impacts the driving style by reducing the car position error and slightly affecting car orientation.

**Table 4.** Performance index for car's position using BPNN with and without GA-PID.

| | Performance Index for Car's Position | | | | |
|---|---|---|---|---|---|
| **Scenario** | **ITAE (m) Driver 1** | **ITAE (m) Driver 2** | **ITAE (m) Driver 3** | **ITAE (m) Driver 4** | **ITAE (m) Driver 5** |
| **Driving behaviour without GA-PID** | 32,980.38 | 10,829.86 | 18,419.31 | 126,787.6 | 45,695.87 |
| **Driving behaviour with GA-PID** | 25,737.86 | 4903.1 | 11,542.43 | 21,467.32 | 18,523.59 |
| **Improvement** | 7242.52 | 5926.76 | 6876.88 | 105,320.3 | 27,172.28 |
| **Percentages** | 21.96% | 54.73% | 37.34% | 83.07% | 59.46% |

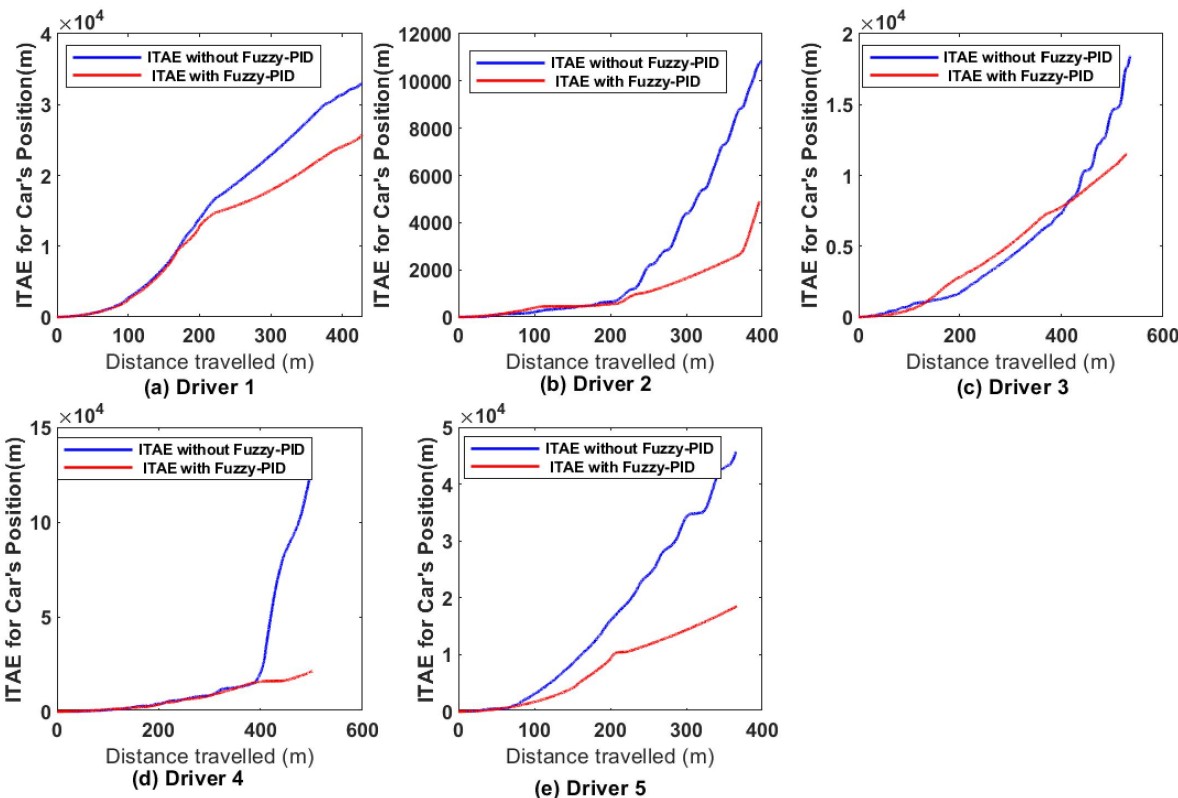

**Figure 15.** ITAE for car's position using BPNN model with and without GA-PID.

5.1.3. ITAE for Car's Orientation Based on GA-PID Controller

Due to individual personalities, different driving behaviour based on orientation was also compensated for by the GA-PID, and this can be seen from the performance index (ITAE) indicated in Figure 16 and the results recorded in Table 5. The ITAE curve of the car's orientation error of various driving behaviours with the GA-PID is under the curve of the car's orientation without the controller for all the drivers after a task. The improvement value produced in Table 5 indicates that the enhancement varies from 1.2% to 7.22% and validates the result displayed in Figure 16. The slight impact on the car orientation error is due to the constant movement of the driver on the steering wheel angle during the driving task.

**Table 5.** Performance index for car's orientation using BPNN with and without GA-PID.

| Scenario | Performance Index for Car's Orientation | | | | |
| --- | --- | --- | --- | --- | --- |
| | ITAE (rad) Driver 1 | ITAE (rad) Driver 2 | ITAE (rad) Driver 3 | ITAE (rad) Driver 4 | ITAE (rad) Driver 5 |
| Driving behaviour without GA-PID | 28,847.46 | 14,056 | 57,108.31 | 48,903.29 | 19,472 |
| Driving behaviour with GA-PID | 22,565.04 | 13,193.2 | 56,420.41 | 46,398.59 | 18,340.5 |
| Improvement | 6282.42 | 862.80 | 687.90 | 2504.70 | 1131.50 |
| Percentages | 22% | 6.14% | 1.20% | 5.12% | 5.81% |

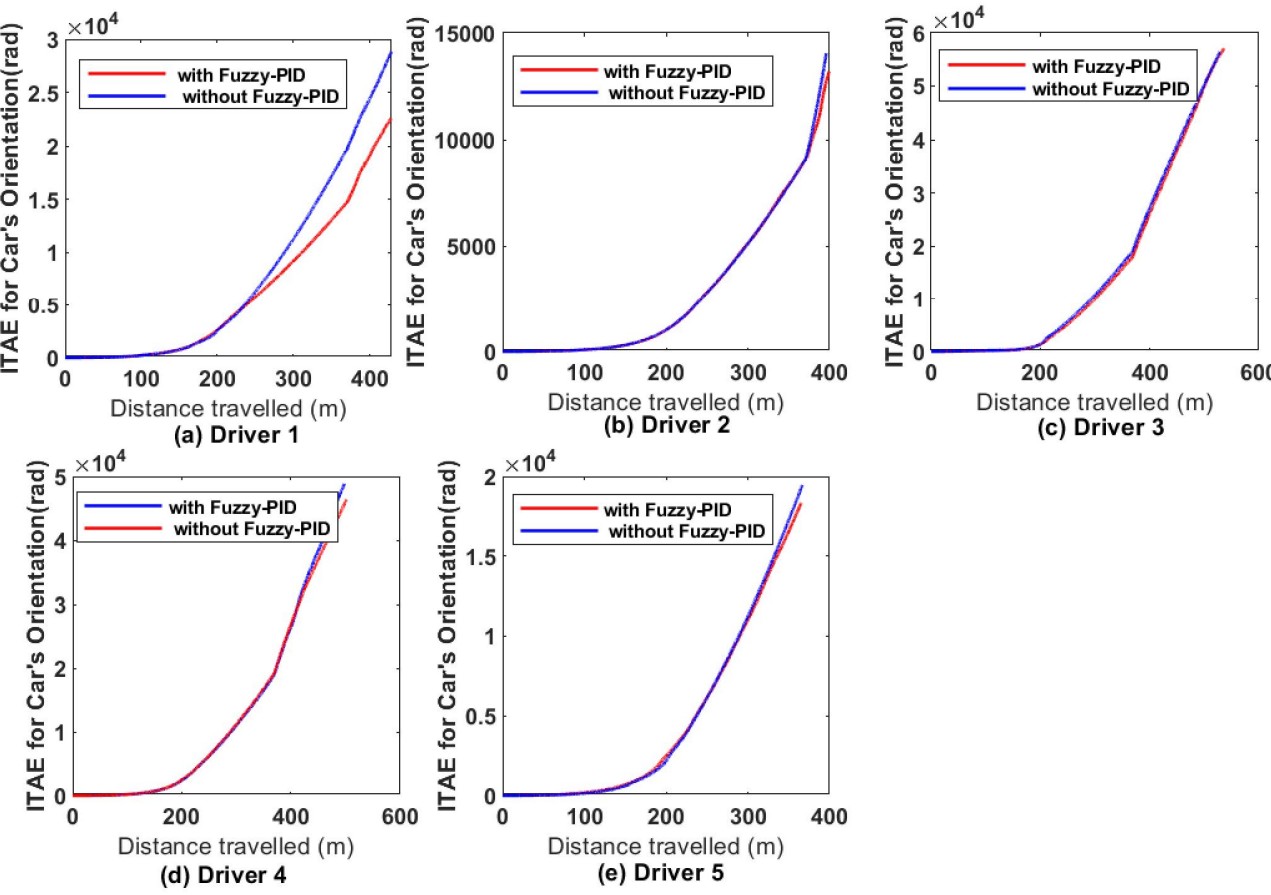

**Figure 16.** ITAE for car's orientation using BPNN model with and without GA-PID.

*5.2. Applied Fuzzy-PID*

After experimenting with the proposed data-driven controller, the graphs representing the PID parameters ($K_p$, $K_i$ and $K_d$) in Figure 17 were generated for each driving behaviour. These results show the variation in the PID parameters for individual driving behaviours. The variation differs from the driver because each driver has a personality or identity. Many variations in driving behaviour generated more change in the PID parameters, which can be seen in drivers 2, 6 and 7. This variation's purpose was to keep the car position on the centerline and reduce the car orientation angle for each driving behaviour. This change in PID gains is due to the fuzzy logic's input membership, which comprises the car position, orientation error and speed following the line, keeping the vehicle in a straight or deviating line.

Figure 18 displays how the optimal fuzzy-PID parameter set is dynamically obtained due to the variation in the car's lateral error, the orientation angle error and the speed for driver one (1) and two (2) after a given task. For the first 100 m for driver one, the car's position error has risen to its peak value at 30 m while the orientation decreased under 0 angle and the speed is 53.8 mile/h. This driving profile allows the PID controller parameter set to change dynamically. Between 100 m and 210 m, the PID gain set is constant because the position error, the orientation angle error and the speed are within the boundary to obtain optimal gains. The orientation angle error drastically changes between 210 m and 225 m, forcing the fuzzy logic to provide the best PID parameter set for this situation. This automatic update of the PID parameters can be observed through the travelled distance. The same interpretation can be applied to driver 2. Let us observe the driving profile and the corresponding PID parameter set between 800 m and 900 m. The parameter set has been changing due to the increase in the car's lateral position error (1.5 m), the decrease in the car's orientation angle error (−3 rad) and the car's speed (54.5 miles/h). These

results demonstrate that the optimal fuzzy-PID parameter changes in different situations to improve the controller performance (flexibility or robustness) by reducing the car's position and orientation angle error for various individual driving styles.

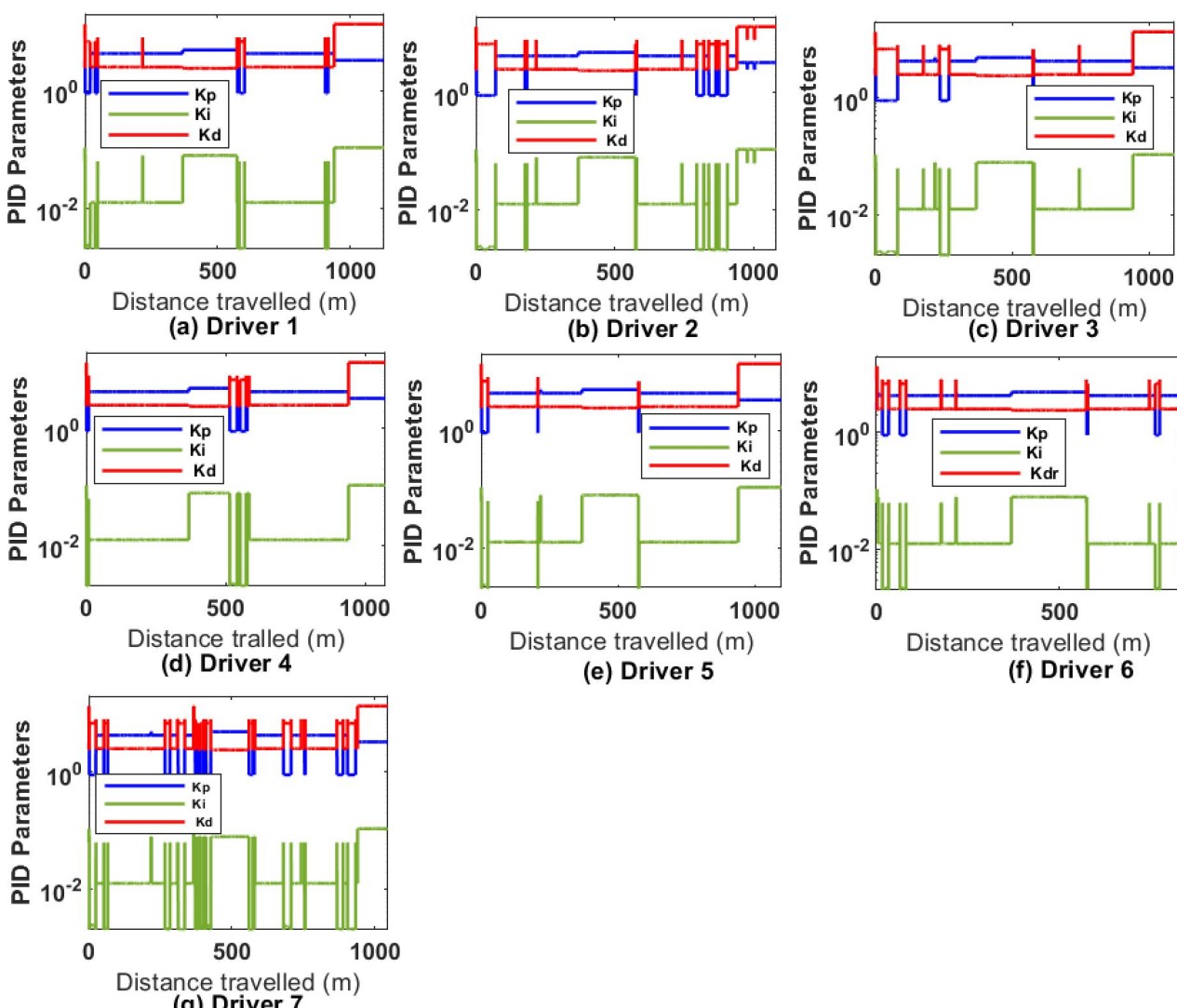

**Figure 17.** Optimal fuzzy-PID parameter set generated from various drivers after a given task.

The speed generated without the fuzzy-PID during the driving shown in Figure 19 indicated a significant variation in individual driving style. The variation is due to the steering wheel's sensitivity, which keeps the car on the centre line because it can easily control a steering wheel at a low speed. The same driving behaviour generated less speed variation with the fuzzy-PID proposed compensator; this means that the obtained fuzzy-PID parameter set was tuned to stabilise the steering wheel angle, keep the car on the centre line and simplify the car yaw angle. The average car's speed provided in Table 6 confirms the statement mentioned above. The average speed results in this table show how the fuzzy-PID controller assisted various drivers appropriately at a high and constant speed compared to the system without.

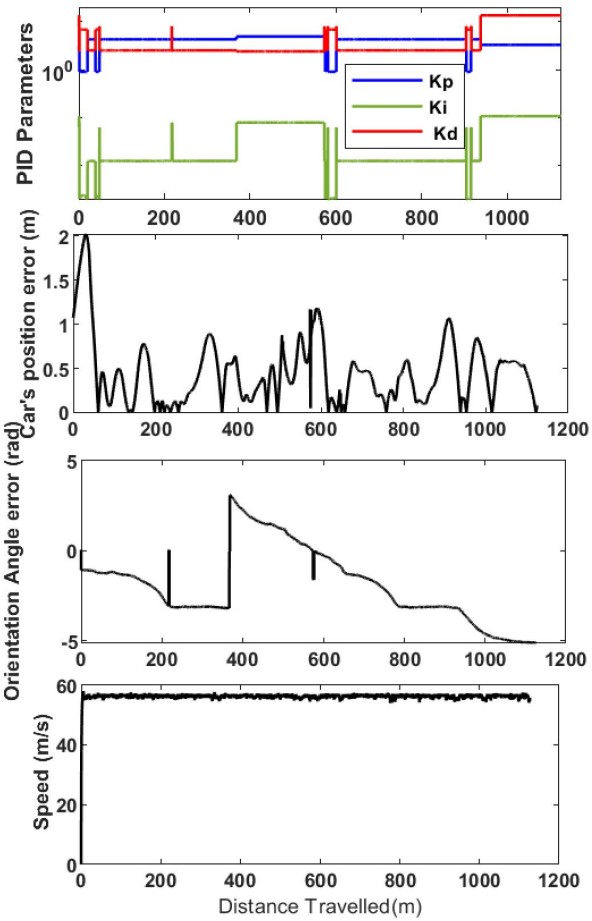

**Figure 18.** Fuzzy-PID parameters change with the car′s position error, orientation angle error and car′s speed in different drivers.

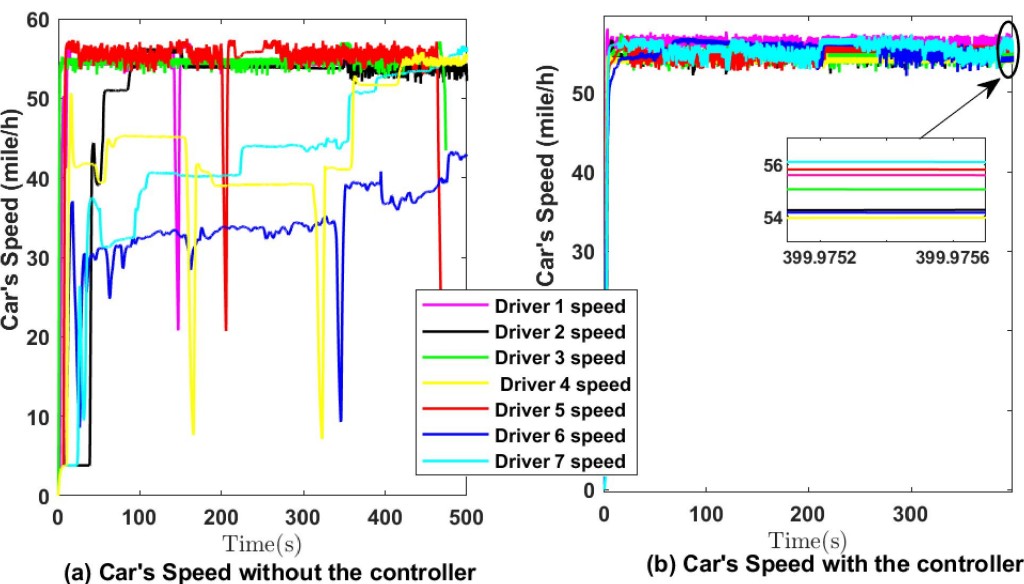

**Figure 19.** Car′s speed without and with controller for the seven drivers.

**Table 6.** Average car's speed (mile/h) with and without fuzzy-PID for various driving behaviours.

| Scenario | Driver 1 | Driver 2 | Driver 3 | Driver 4 | Driver 5 | Driver 6 | Driver 7 |
|---|---|---|---|---|---|---|---|
| **Driving Speed with Fuzzy-PID** | 56.1204 | 54.4615 | 53.9873 | 53.8683 | 54.7353 | 55.0315 | 55.0937 |
| **Driving Speed without Fuzzy-PID** | 53.853 | 54.5193 | 49.5171 | 43.0751 | 52.6064 | 33.3196 | 41.7989 |

*5.3. Driving Results*

Figure 20 below represents the seven driving behaviours for a specific pathway in two different situations. The results show how the fuzzy-PID compensator assisted the driver in getting closer to the desired centre line in a straight line or when the car is subjected to a turn.

Figure 21 presents the curves representing the car's lateral displacement error for seven driving behaviours with the data-driven controller (fuzzy-PID) and without in a function of the distance travelled by following a centre line. This result stipulates that the curve which illustrates the car's lateral displacement error is higher for the driving behaviour without the compensator for all subjects (drivers). Based on the performance index of any task, the lower the error, the better the controller. The data-driven controller (fuzzy-PID), based on its fuzzy rules and input and output fuzzy subset, tuned the PID parameter set to adjust the car's position close to the reference, which is the centre line for the seven drivers. This difference in error corroborates with the result displayed in Figure 20.

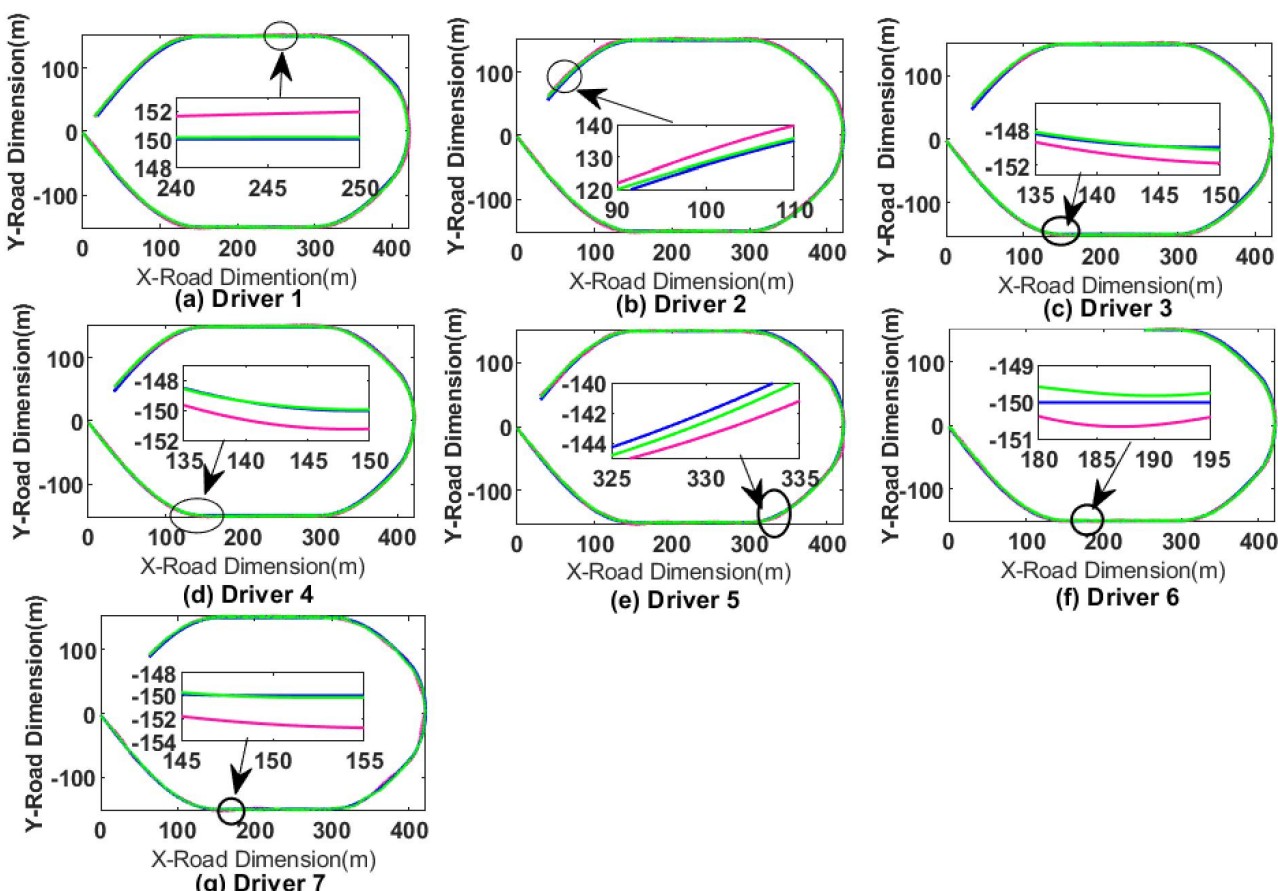

**Figure 20.** Seven driving behaviours with and without the fuzzy-PID controller after a given task (blue (-), magenta (-) and green (-) colours represent, respectively, the driving pathway, driving behaviour without fuzzy-PID and driving behaviour with fuzzy-PID).

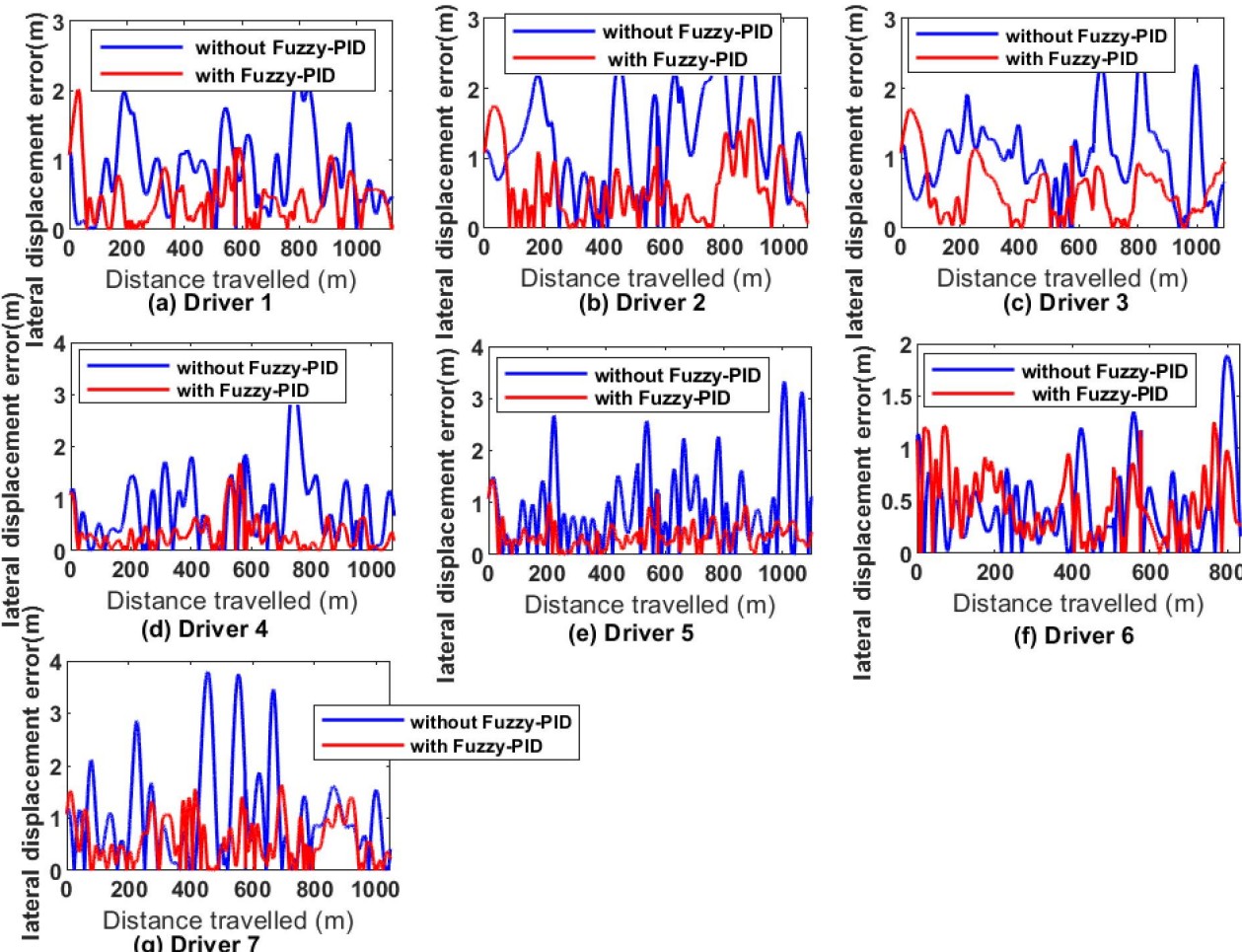

**Figure 21.** Car's position error with and without the fuzzy-PID controller after a travelled distance (blue (-) and red (-) represent, respectively, the error without fuzzy-PID and with fuzzy-PID).

*5.4. Integral Time Absolute Error (ITAE) for Car Position*

The outcome displayed in Figure 22 indicates the curve of the performance index (ITAE) of seven driving behaviours in car lateral displacement between a system with fuzzy-PID and the system without for a given distance. The curve representing the performance index (ITAE) with fuzzy-PID is low, unlike the curve without the seven driving behaviours. These results show how the performance index is lower for the task duration of the driving behaviour combined with the fuzzy-PID. It is essential to highlight that low ITAE is observed for drivers 1 to 7.

Table 7 compares the total integral time absolute error of the car position between the driving behaviour with fuzzy-PID and without for a given job. This table confirms the curve presented in Figure 22. The indicated values demonstrate that the driving behaviour error due to the car position and the desired pathway is significantly low for all the drivers with the compensator (fuzzy-PID). The improvement recorded varies between 56% and 77%. The progress noticed in each driving behaviour reveals the fuzzy-PID controller's robustness for this line-keeping scenario.

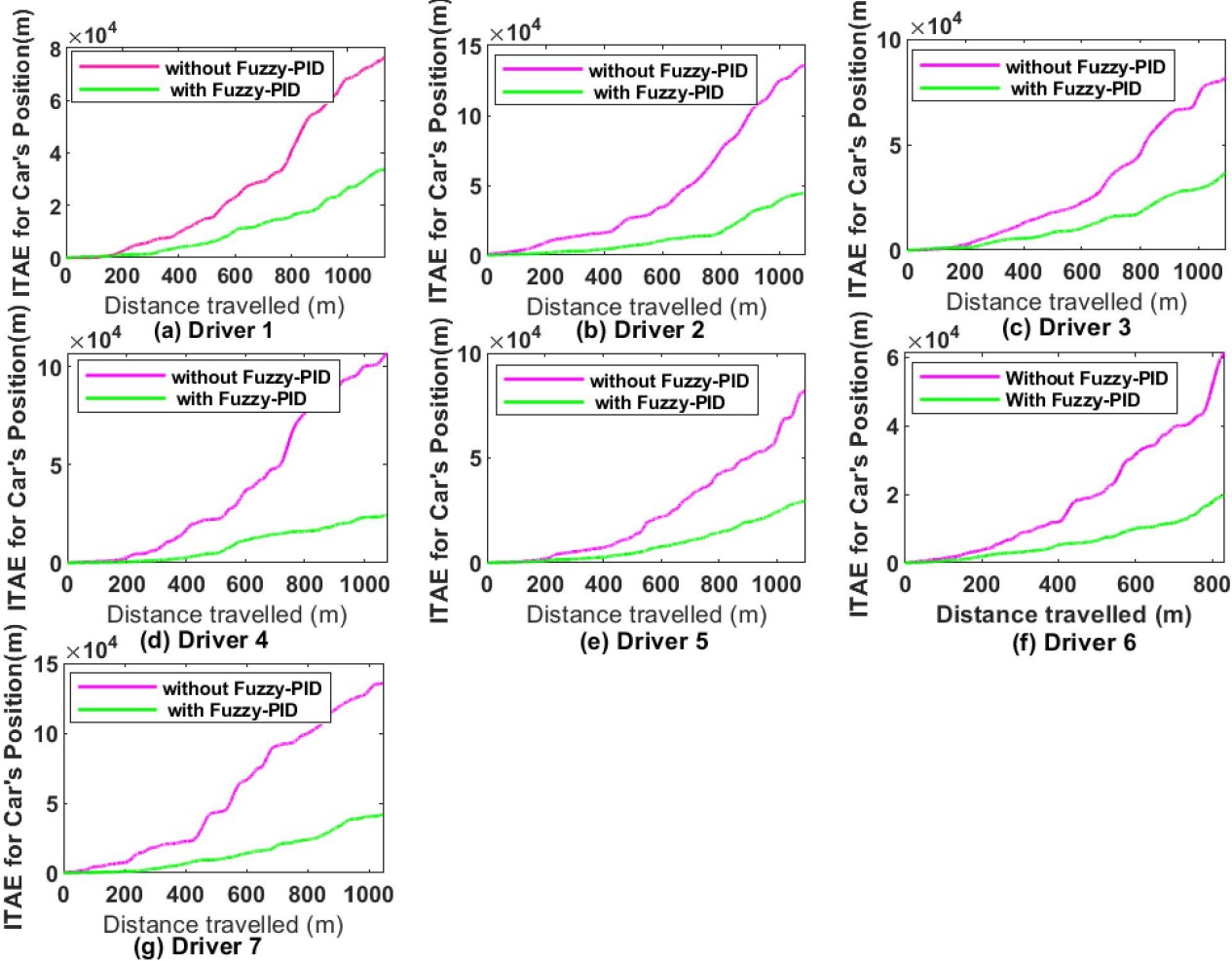

**Figure 22.** ITAE for car's position with and without fuzzy-PID controller.

**Table 7.** Car's position performance index with and without the fuzzy-PID controller.

| Scenario | Performance Index for Car Position | | | | | | |
|---|---|---|---|---|---|---|---|
| | ITAE (m) Driver 1 | ITAE (m) Driver 2 | ITAE (m) Driver 3 | ITAE (m) Driver 4 | ITAE (m) Driver 5 | ITAE (m) Driver 6 | ITAE (m) Driver 7 |
| **Driving behaviour without fuzzy-PID** | 76,150.91 | 136,021.6 | 81,759.06 | 106,950.6 | 82,891.66 | 61,278.152 | 135,595.3 |
| **Driving behaviour with fuzzy-PID** | 33,634.73 | 44,330.49 | 36,789.21 | 24,357.55 | 29,514.59 | 19,682.56 | 41,600.55 |
| **Improvement** | 42,516.18 | 91,691.07 | 44,969.85 | 82,593.03 | 53,377.07 | 41,595.592 | 93,994.76 |
| **Percentages** | 56% | 67% | 55% | 77% | 64% | 68% | 69% |

*5.5. Integral Time Absolute Error (ITAE) for Car Orientation*

Figure 23 and Table 8 show how the performance index (ITAE) of the car orientation for various driving behaviours has been affected by the fuzzy-PID controller for a given task. The curves and Table 8 indicate the improvement in the driving behaviour for car orientation for all the drivers. This improvement is low for some drivers while significant for others. The enhancement fluctuates between 6.1% and 63%.

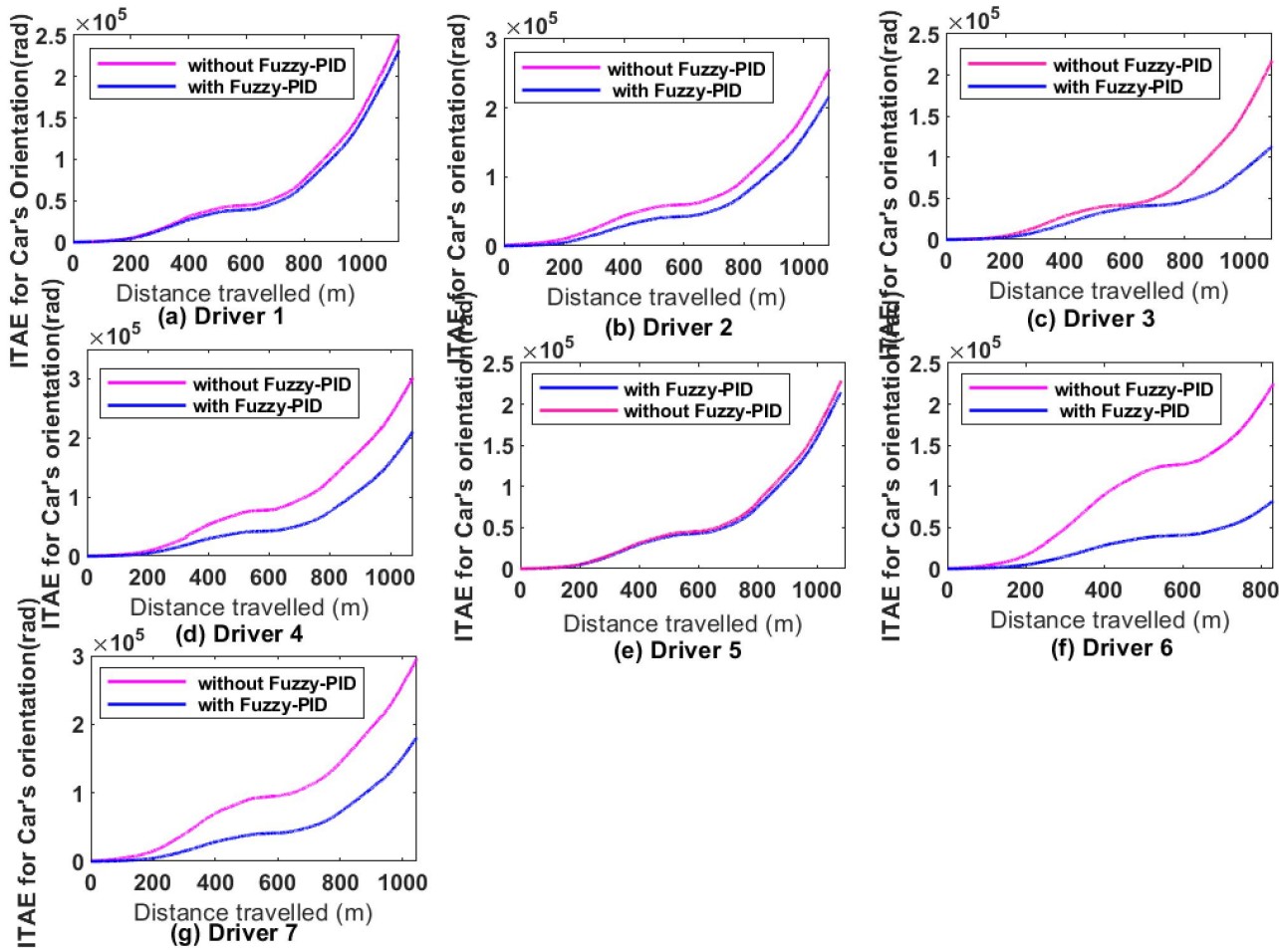

**Figure 23.** ITAE for car's orientation with and without fuzzy-PID controller.

**Table 8.** Car's orientation performance index with and without the fuzzy-PID controller.

| Scenario | Performance Index for Car Orientation | | | | | | |
|---|---|---|---|---|---|---|---|
| | ITAE (rad) Driver 1 | ITAE (rad) Driver 2 | ITAE (rad) Driver 3 | ITAE (rad) Driver 4 | ITAE (rad) Driver 5 | ITAE (rad) Driver 6 | ITAE (rad) Driver 7 |
| Driving behaviour without fuzzy-PID | 248,431.6 | 258,389.1 | 219,338.83 | 304,274 | 228,586 | 225,661.2 | 295,849.9 |
| Driving behaviour with fuzzy-PID | 231,439 | 216,183.4 | 113,396.1 | 210,842.6 | 214,620.7 | 82,985.34 | 181,997 |
| Improvement | 16,992.59 | 42,205.66 | 105,942.73 | 93,431.33 | 13,965.30 | 142,675.88 | 113,852.9 |
| Percentages | 7% | 16% | 48.30% | 30.71% | 6% | 63% | 38% |

### 5.6. Haptic Feedback Torque

Figure 24 compares the driver's use of the haptic feedback torque on the steering wheel in two situations (with and without the fuzzy-PID). The driver with higher performance or driving with the proposed controller associated shows less assistive feedback torque, unlike the system without. This can be proven by looking at how often the haptic force feedback curve keeps on the origin line of the X-axis because, at the origin, the torque is zero (0), meaning that the drivers do not need any assistance.

Table 9 indicates the total torque needed for each driving behaviour after a complete task. The values are small with the proposed fuzzy-PID compared to the system without this controller.

**Table 9.** Haptic feedback steering wheel torque (Nm) compared with and without the fuzzy-PID controller.

| Scenario | Haptic Feedback Steering Wheel Torque (Nm) | | | | | | |
|---|---|---|---|---|---|---|---|
| | Driver 1 | Driver 2 | Driver 3 | Driver 4 | Driver 5 | Driver 6 | Driver 7 |
| Feedback Torque without Fuzzy-PID | 1485.2 | 1816.7 | 1665.2 | 1485.6 | 1677.2 | 1149.7 | 1911 |
| Feedback Torque with Fuzzy-PID | 890.35 | 1006.4 | 980.75 | 604.4 | 959.9 | 819.1 | 1217.4 |
| Improvement | 594.85 | 810.3 | 684.45 | 881.2 | 717.3 | 330.6 | 693.6 |
| Percentages | 40% | 45% | 41% | 59% | 43% | 29% | 36% |

Reducing the assistive feedback torque on the steering wheel diminishes the conflict between the driver and the haptic feedback system. Furthermore, the after-effect during continuous haptic feedback torque is also reduced.

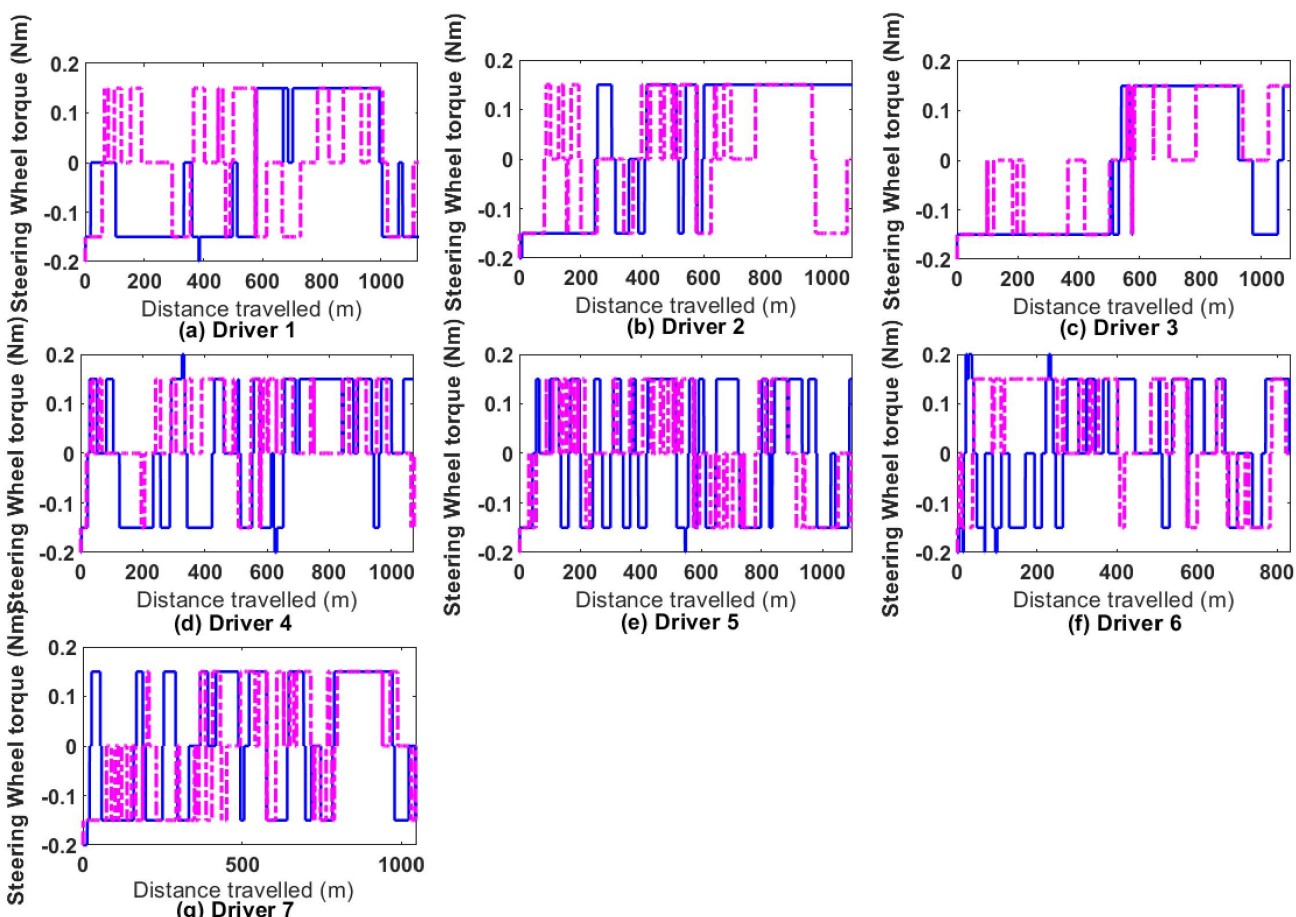

**Figure 24.** Applied haptic feedback steering wheel torque with and without the fuzzy-PID controller on various drivers (blue (-) and magenta (-) colours represent the torque without controller and with controller, respectively).

The outcome presented in Figure 25 confirms that the bandwidth haptic feedback guidance provided the assistive steering wheel torque when the predicted car's lateral displacement error exceeded 0.3 m with reference to the lane centre. The amount of haptic assistive torque depends on the car's position, the error bandwidth and the driving speed. These results show that, when the car's lateral error exceeds the following interval [0.3 m −0.3 m] and the car's speed is lower than 30 mile/h, the bandwidth assistive guidance provided a torque of ±0.2 N·m, and when the speed exceeded 30 mile/h, the assistive steering wheel torque was ±0.15 N·m. This force difference is because the steering wheel angle is too sensitive at high velocity. In addition, when the car's position is within the mentioned interval, the drivers are not assisted. The haptic force feedback with the controller

shows that the driver drove within the required interval in many situations compared to the scenario without the controller. The curve representing the torque feedback with the fuzzy-PID controller stays more at the zero (0) axis, unlike the curve without.

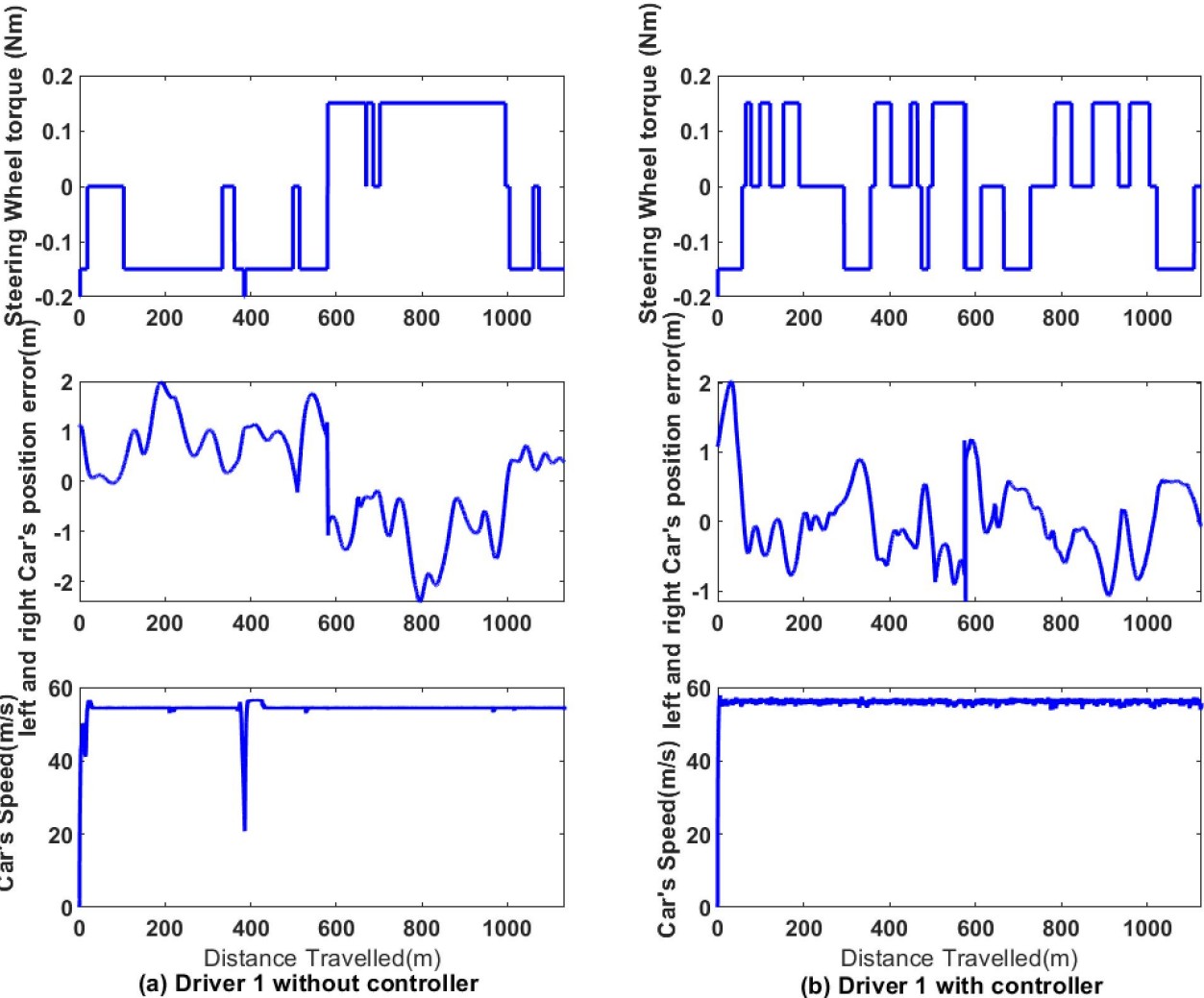

**Figure 25.** Haptic feedback torque change with error, speed, and with and without the fuzzy-PID controller.

*5.7. Performance Analysis between the Proposed Controller and the GA-PID*

The purpose of this section is to demonstrate the performance of the new controller against any existing one. A third experiment was conducted; the GA-PID parameters employed were processed in Section 5.1 and the values indicated in Table 3. The results obtained with different controllers indicated their car position, orientation angle performances and the driver's comfort.

The dynamically tuned fuzzy-PID parameters ($K_p$, $K_i$, $K_d$) are displayed in Figure 26. The change in the curve during the driving task is based on the car's speed, position and orientation angle error. The controller performance is indicated in Figure 26.

Figure 27 represents the lateral displacement error for five drivers in three situations. The curves show that the driving scenario with a fuzzy-PID controller is lower than the GA-PID and the scenario without for all the drivers. These results clearly demonstrate the performance and robustness of the proposed controller.

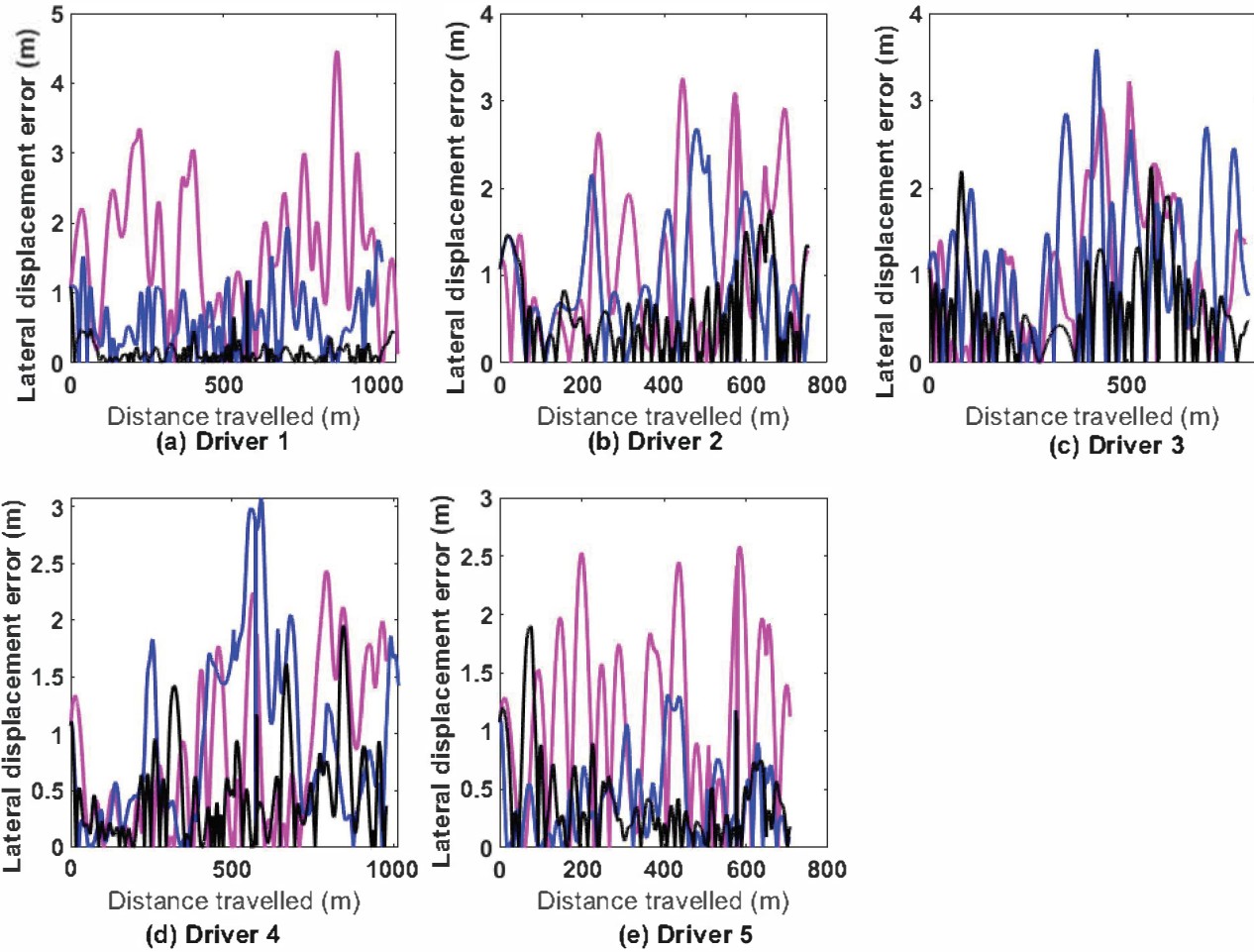

**Figure 26.** Comparison of controller in car's displacement error with fuzzy-PID and GA-PID controller (magenta (-), blue (-) and black (-) colours represent, respectively, the error without controller, with GA-PID and with fuzzy-PID).

5.7.1. ITAE between the Proposed Controller and the GA-PID for Car Position

Integral time absolute error is the method used to analyse the performance in the data-driven controller; Figure 28 and the improved values in Table 10 below indicate that the GA-PID controller performs worse in the car lateral displacement angle error than the proposed controller (fuzzy-PID). The efficiency of the new controller is due to the fact that the PID controller is dynamically tuned based on the variation in the car position, orientation angle error and speed.

**Table 10.** ITAE performance of the proposed controller and GA-PID.

| Scenario | Performance Index for Car's Position | | | | |
|---|---|---|---|---|---|
| | ITAE (m) Driver 1 | ITAE (m) Driver 2 | ITAE (m) Driver 3 | ITAE (m) Driver 4 | ITAE (m) Driver 5 |
| Driving behaviour without controller | 89,178 | $1.58 \times 10^5$ | 94,400 | $1.22 \times 10^5$ | $6.73 \times 10^4$ |
| Driving behaviour with GA-PID | 53,325 | 66,207 | 51,320 | 82,845 | 66,207 |
| Improvement with GA-PID | 35,853 | 91,493 | 43,080 | 39,085 | 1112 |
| Percentages | 40% | 58% | 46% | 32% | 2% |
| Driving behaviour with fuzzy-PID | 10,183 | 19,918 | 28,220 | 21,467.32 | 34,420 |
| Improvement with fuzzy-PID | 78,995 | $1.38 \times 10^5$ | 66,180 | 100,462.7 | 32,899 |
| Percentages | 88.58% | 87.37% | 70.11% | 82.39% | 48.87% |

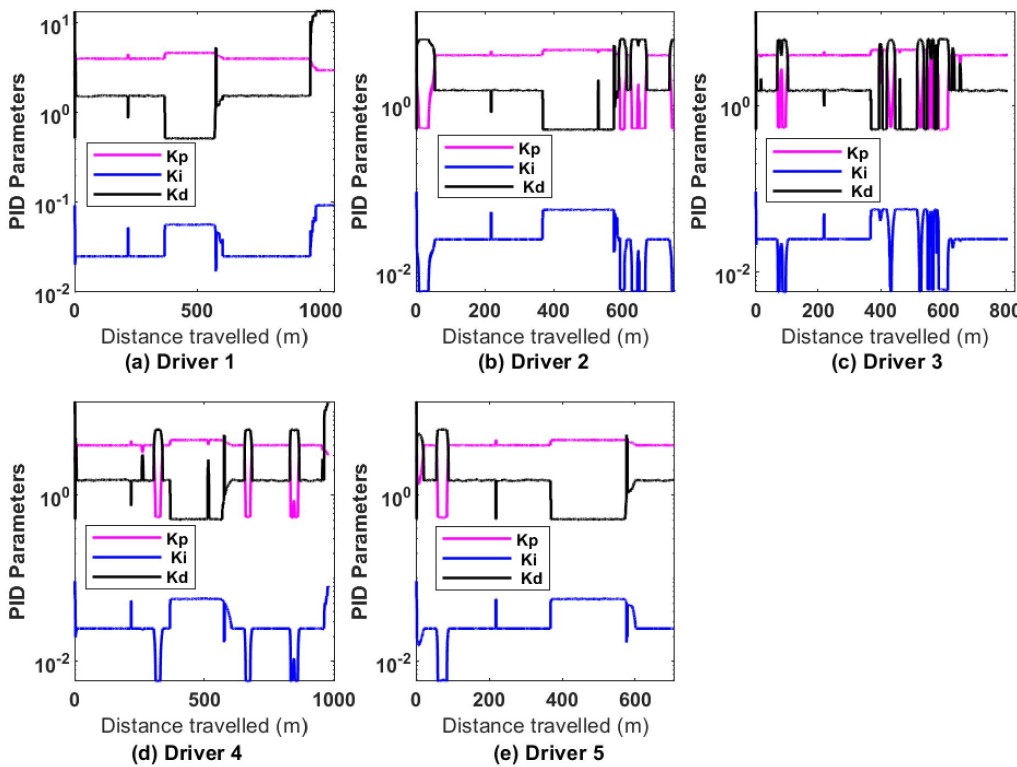

**Figure 27.** Optimal fuzzy-PID parameter set generated from five drivers after a given task.

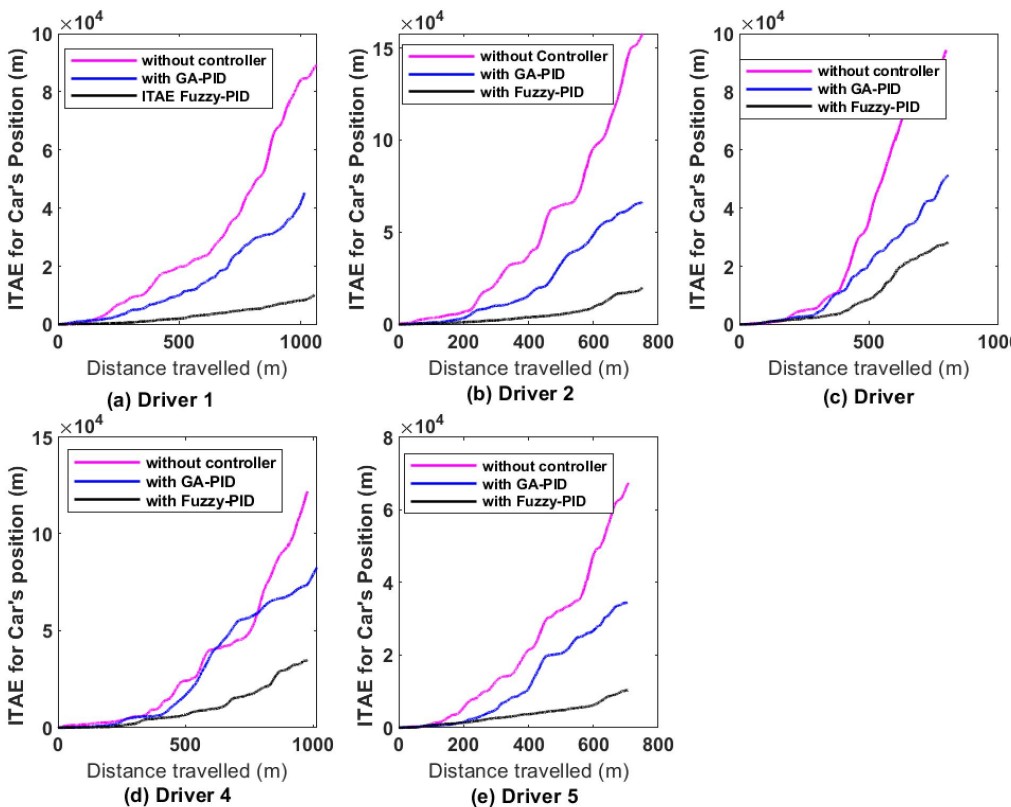

**Figure 28.** ITAE comparison of controller output in car's position .

### 5.7.2. ITAE between the Proposed Controller and the GA-PID for Car Orientation

The curve displayed in Figure 29 and the improvement values in Table 11 show that the GA-PID has less impact on car orientation than the proposed controller. Based on

the table, the GA-PID controller did not impact driver 5, while the proposed controller positively impacted all the drivers.

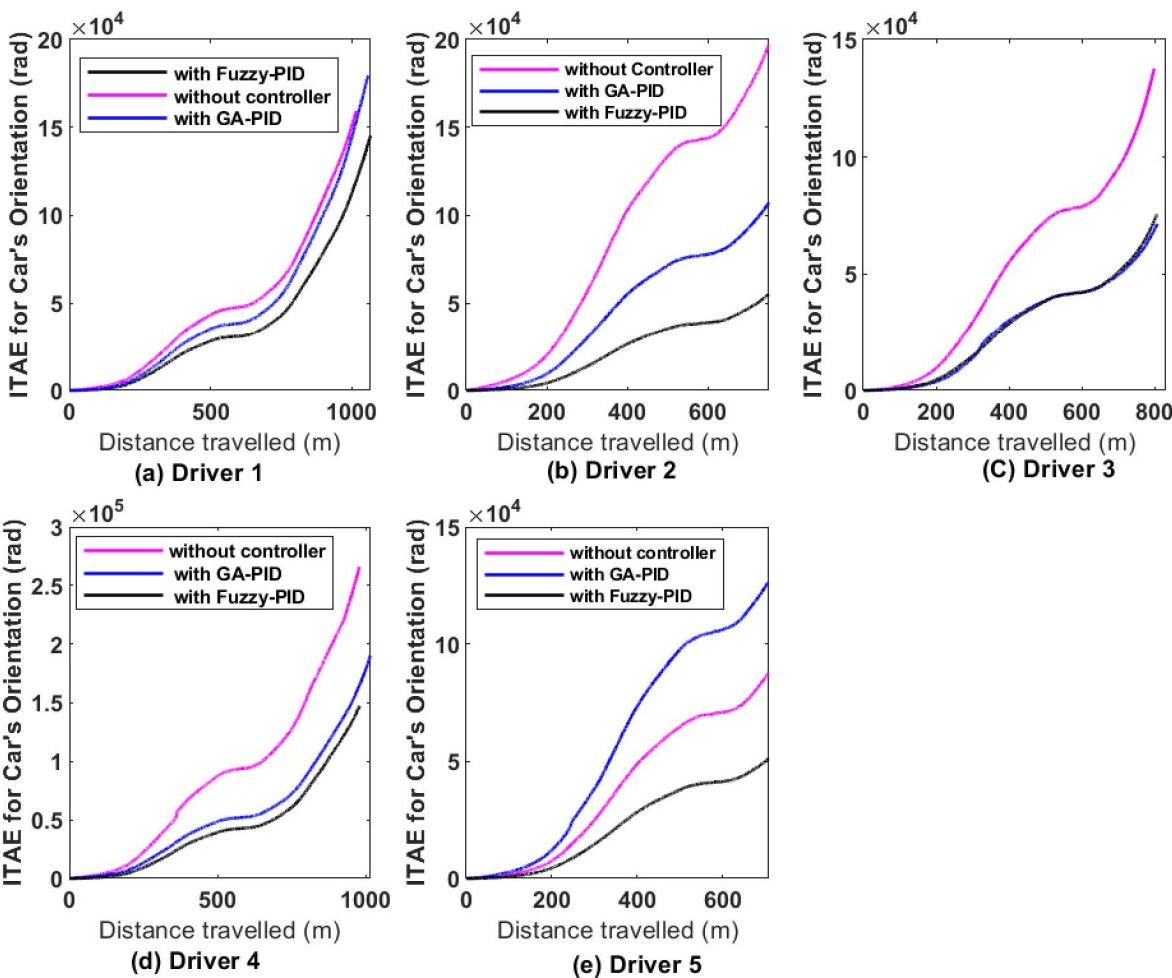

**Figure 29.** ITAE comparison of the controller outcome for car's orientation.

**Table 11.** ITAE comparison of the GA-PID and the proposed controller.

| Scenario | Performance Index for Car's Orientation | | | | |
| --- | --- | --- | --- | --- | --- |
| | ITAE (rad) Driver 1 | ITAE (rad) Driver 2 | ITAE (rad) Driver 3 | ITAE (rad) Driver 4 | ITAE (rad) Driver 5 |
| Driving behaviour without controller | $2.05 \times 10^5$ | $1.98 \times 10^5$ | $1.38 \times 10^5$ | $2.66 \times 10^5$ | $8.77 \times 10^4$ |
| Driving behaviour with GA-PID | $1.79 \times 10^5$ | $1.07 \times 10^5$ | $7.15 \times 10^4$ | $1.96 \times 10^5$ | $1.26 \times 10^5$ |
| Improvement with GA-PID | 25,810 | 90,267 | 66,112 | 69,900 | −38,517 |
| Percentages | 13% | 46% | 48% | 26% | −44% |
| Driving behaviour with fuzzy-PID | $1.45 \times 10^5$ | 55,322 | 75,620 | $1.48 \times 10^5$ | 51,208 |
| Improvement with fuzzy-PID | 59,930 | $1.42 \times 10^5$ | 61,990 | 118,290 | 36,525 |
| Percentages | 29.19% | 72.02% | 45.05% | 44.48% | 41.63% |

### 5.7.3. Haptic Feedback Torque with the Proposed Controller and the GA-PID

The values in Table 12 and the curves in Figure 30 show that both controllers affect the feedback torque, but the proposed controller affects it more. The smaller the haptic feedback torque, the higher the driver's comfort in operating the car without assistance.

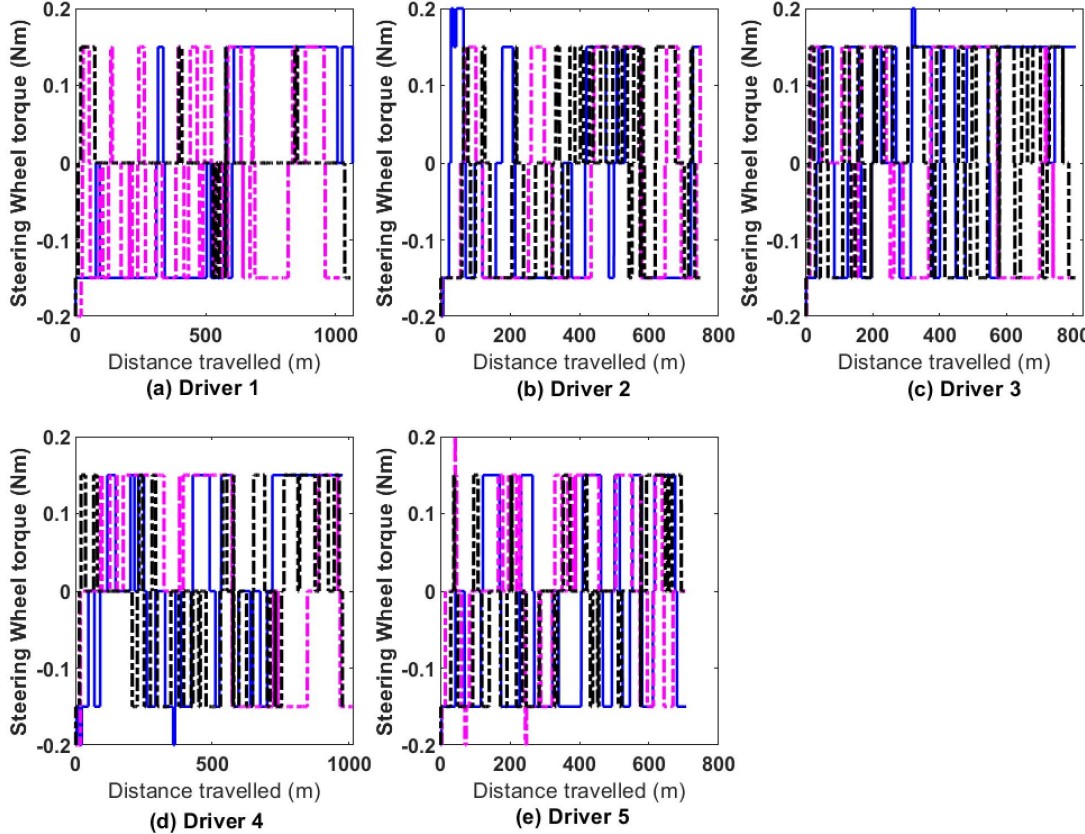

**Figure 30.** Comparison of controllers in haptic feedback torque (magenta (-), blue (-) and black (-) colours represent, respectively, the torque without controller, with GA-PID and with fuzzy-PID).

The graph shows the fuzzy-PID's performance compared to GA-PID; the longer the curve stays on zero, the greater the driver's comfort; when the stays on zero, it means that there is no driving assistance (haptic force).

**Table 12.** Haptic feedback steering wheel torque (Nm) comparison.

| Scenario | Haptic Feedback Steering Wheel Torque (Nm) | | | | |
| --- | --- | --- | --- | --- | --- |
| | Driver 1 | Driver 2 | Driver 3 | Driver 4 | Driver 5 |
| Driving behaviour without controller | 1249.10 | 1733.50 | 1173.60 | 1442.80 | 1171.50 |
| Driving behaviour with GA-PID | 1156.90 | 1207.70 | 988.35 | 1137.00 | 811.45 |
| Improvement with GA-PID | 92.20 | 525.80 | 185.25 | 305.80 | 360.05 |
| Percentages | 7.4% | 30.3% | 15.8% | 21.2% | 30.7% |
| Driving behaviour with fuzzy-PID | 204.10 | 767.25 | 969.60 | 758.15 | 534.00 |
| Improvement with fuzzy-PID | 1045 | 966.25 | 204 | 684.65 | 637.5 |
| Percentages | 83.66% | 55.74% | 17.38% | 47.45% | 54.42% |

## 6. Discussion

In this study, the proposed driving behaviour controller was designed. This fuzzy-PID controller combined the fuzzy logic technique with the PID controller, which considered the non-linearity of human behaviour, individual driving styles and the complexity of the human–car–road system to minimise the car position and orientation errors following a centre line, unlike a system without. This section of the paper will analyse and highlight the performance of the proposed model-free driving behaviour controller with a haptic force feedback system through several results obtained after conducting experiments.

### 6.1. Human Driving Behaviour Model Based on BPNN Performance Analysis

Due to the non-linearity of the driving behaviour, which can be caused by multiple factors such as unique driving style, weather conditions and road structure, the BPNN was used as the driving behaviour model architecture. The driving simulator data was collected, processed, trained, tested and validated. The results displayed in Figure 12 show the performance of the BPNN plot for various driving behaviours where the training, validation and testing curve performance (MSE) decrease with the epoch number. Table 2 illustrates that the mean squared error was low (from 0.14 to 0.37), the regression value was between 80.3% and 93%, and the epochs fluctuated between 100 and 520. These results show the flexibility of the driving behaviour BPNN because it handled multiple driving input characteristics (position and orientation angle error) and output actions (steering wheel angle and car's acceleration). In addition, it has shown its adaptability to non-linear driving behaviour or dynamic systems, where its weight was well updated for a change in situation to improve driving behaviour. The driving behaviour BPNN model presented in Figure 13 demonstrates how much the model was closer to the driving simulator pathway; this result was also due to the data processing (data normalisation and denormalisation).

### 6.2. GA-PID Controller with BPNN Model Performance Examination Due to the ITAE Criterion

In this study, the driving behaviour model was mapped by a BPNN and the data-driven controller was chosen. A GA-PID controller was used and the genetic algorithm approach found the optimal PID controller gains offline to avoid manual tuning. In the process of this controller, the ITAE was used as the fitness function (combined car's position error and orientation angle error) and the performance index. The GA was used to search for the best PID controller gains that minimised the ITAE index. Table 2 presents a different optimal PID parameter set obtained by the GA for a given travelled distance, and Figures 15 and 16 show how low the ITAE criterion for a car's lateral displacement and orientation is with a system with GA-PID compare to a system without; this observation can also be seen in Tables 2 and 3 where the improvement is noticed (22% to 83% for the car's position and 1.2% to 21.8% for the car's orientation). It can be concluded that the GA provides a low ITAE index for all driving behaviours based on the car's position and orientation angle, which shows evidence of the performance of the GA-PID controller to get closer to the desired pathway.

### 6.3. Robustness and Transparency Analysis of Fuzzy-PID Controller Due to the ITAE Performance Criteria

Based on the individual driver's personality, human inaccuracy and uncertainties during a given driving task, the fuzzy-PID controller used the fuzzy logic approach to provide the best PID gains set that improved the performance of the controller by monitoring various driving profiles such as the car's lateral displacement error, car's orientation error and the speed, which create uncertain operating conditions for each driver. Based on the current driving task of each driver, the fuzzy logic section adjusted the parameter set ($K_p$, $K_i$, $K_d$) of the PID controller dynamically, which directly impacted the steering wheel angle and, therefore, enhanced the performance of the controller by reducing the car's lateral error and car's orientation angle. The results indicated in Figure 18 show that the PID parameter set is changing dynamically based on the change in driving profile, such as speed, variation in car's lateral error and car orientation angle error for a travelled distance. The results indicated in Figures 22 and 23 and Tables 6 and 7 also reveal the robustness and the adaptability of the fuzzy-PID controller, meaning its ability to encounter different driving behaviours for various drivers. Figure 19 illustrates the unstable driving speed of a system without a controller. This variation is due to the instability of the steering wheel angle; the same graph shows various constant speeds with the system with the fuzzy-PID controller. Table 6 shows that the fuzzy-PID controller impacted the driving speed. The average car's speed is low (33.3 miles/h to 54 miles/h) for driving behaviour without a controller and high (53.7 miles/h to 56.12 miles/h). These pieces of information

affirm that the fuzzy-PID controller improves the car's lateral position or orientation angle, and enhances the driving ability by allowing the individual to use a high and constant speed when keeping the vehicle on a centre line for a given task.

### 6.4. Fuzzy-PID Controller Performance on Haptic Feedback Torque

The haptic feedback torque results also point out that the drivers were not assisted when the lateral error was within the bandwidth ([0.3 m −0.3 m]); Figure 25 and Table 9 demonstrate that the driving behaviour with the fuzzy-PID controller drove for some distance without the haptic feedback assistance. The values shown in Table 9 affirmed that the proposed controller had improved the driving skill of all the drivers because the total amount of torque recorded for a travelled distance with the system without the fuzzy-PID controller is higher, unlike the total amount of driving behaviour with the proposed controller which is lower. This table also reveals that the improvement in driving without assistive torque varies from 29% to 59.32%, which means that the proposed data-driven fuzzy-PID controller has significantly reduced the drivers' after-effect caused by the continuous haptic guidance.

### 6.5. Performance Comparison between the Proposed Controller and the GA-PID Controller

The controller performance index (ITAE) for the driving behaviour in car position and orientation shows that the proposed data-driven controller is better than the GA-PID controller. The improvement rate in Table 10 indicates that the system with the GA-PID controller improved the car's position error from 2% to 58% while the fuzzy-PID enhanced it from 48.87% to 88.58% for the five drivers. The improvement is also noticed in the car orientation angle error shown in Table 11 where 21% to 72% is observed for the proposed controller and less for the GA-PID. It is noticed that, in driver five, the GA-PID controller does not impact the car orientation angle error. Furthermore, the results in Table 12 demonstrate how the proposed controller allows the driver to achieve a task with less haptic torque assistance. The improvement rate varies from 17.38% to 83.66% for the system with a fuzzy-PID controller, whereas 7.4% is registered for the system with a GA-PID controller. These results on the five driving behaviours validated the proposed controller's performance and robustness.

In conclusion, all the procedures and performances obtained in this work, such as the car's lateral position, orientation angle error improvement, the driving speed enhancement and the low assistive haptic feedback force, exhibit that various driving behaviours of different drivers and changed situations were controlled by designing a compelling combined fuzzy logic and PID controller parameter set. The fuzzy logic input and output profiles were determined by mapping the driving behaviour based on BPNN; then, the BPNN model and the GA-PID were used to find the best fitness function that generated the optimal PID parameter set.

## 7. Conclusions

This paper presented a proposed model-free fuzzy-PID controller, which addressed the challenges of minimising different driving behaviour errors caused by the individual driving style (car's position, orientation angle and speed) and the road geometric situation (straight and corner road). This proposed controller combined the fuzzy logic technique with the PID controller, which did not need a mathematical model. The genetic algorithm obtained the optimal PID gains set utilised in the proposed fuzzy-PID controller, which minimised the simultaneous loss function ITAE of the car's position and orientation error offline. This dynamic change in the PID parameter set in different driving situations or styles adjusted the steering wheel angle, allowing the car's position and orientation angle to get closer to the desired pathway.

It was noticed that the stationary lateral displacement error varies or is never zero during a driving task because the PID controller in this study does not form a closed loop on its own and the driver movement considered as human in the loop is not always constant

when driving. Additionally, the proposed controller aims not to remove the stationary error but to assist the driver in easily manoeuvring the steering wheel for diverse tasks.

The discussion and the experimental results analysis of the proposed controller revealed that the performance index ITAE of the car's lateral displacement error and ITAE for the car's orientation angle error for individual driving behaviours was reduced significantly; the lower ITAE, the better the controller performance.

Compared to an existing GA-PID, the proposed fuzzy-PID controller performed better in lowering the car position performance index for five drivers. The outcome from Table 10 shows that, although the GA-PID ameliorated the driving behaviour from 2% to 58%, the proposed controller presents a significant enhancement, which varies from 48.87% to 88.58%. These performance differences are also observed in the car's orientation. Table 11 revealed a poor performance in drivers where no improvement was noted.

Furthermore, the proposed controller contributed to the comfort of various drivers, and this can be demonstrated in Figures 24 and 30 and Tables 9 and 12. The small influence of haptic steering wheel feedback torque (low after-effect) on drivers and the ability of each driving behaviour performance in different operating situations consolidated the performance of the fuzzy-PID controller to handle various driving styles. Additionally, the capability of multiple drivers to manoeuvre the steering wheel at high speed with the proposed data-driven controller is indicated in Figure 19 and Table 6. The simulation and experiment outcomes have validated the proposed fuzzy-PID controller's flexibility, effectiveness and robustness in driving behaviour.

However, future research needs to enhance the proposed controller by further improving driving behaviour errors by considering road conditions (dryness and ice). In addition, a further study on a new controller that will feel the conflict between the haptic steering feedback torque and the driver will improve the driver's comfort.

**Author Contributions:** Conceptualisation, S.D., S.I.N.T. and K.D.; methodology, S.I.N.T. and S.D.; validation, S.D., K.D., S.I.N.T. and Q.L.; formal analysis, S.I.N.T.; resources, S.D., S.I.N.T., K.D. and Q.L.; writing—original draft preparation, S.I.N.T.; writing—review and editing, S.D., S.I.N.T., K.D. and Q.L.; supervision, S.D. and K.D. All authors have read and agreed to the published version of the manuscript.

**Funding:** This work is founded on the research supported in part by the National Research Foundation of South Africa (Grant Numbers SRUG2203291049 and 145975), Kunming University Foundation (No. YJL2205), and the Foundation of Yunnan Province Science and Technology Department (No.202305AO350007).

**Institutional Review Board Statement:** The study was conducted in accordance with the Declaration 642 of Helsinki and approved by the Faculty Research Ethics Committee (FREC) of Tshwane University 643 of Technology (protocol code FCRE2021/07/026-EBE, 14-09-2021).

**Informed Consent Statement:** Informed consent was obtained from all subjects involved in the 645 study.

**Data Availability Statement:** Data are contained within the article.

**Acknowledgments:** Our gratitude and appreciation go to the National Research Foundation, French South Africa Institute of Technology (F'SATI), and all members and colleagues of the Tshwane University of Technology (TUT), especially the Department of Electrical Engineering, for providing the facilities and material to conduct this research.

**Conflicts of Interest:** The authors declare no conflicts of interest.

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
