# Peer review of "Data-Driven Controller for Drivers’ Steering-Wheel Operating Behaviour in Haptic Assistive Driving System"

_electronics, doi:10.3390/electronics13061157_

Round 1

Reviewer 1 Report

Comments and Suggestions for Authors

The following paper introduces a Fuzzy-PID controller designed to reduce driving errors resulting from individual driving styles. While acknowledging the authors' evident dedication to the research, I believe that this manuscript could benefit from some improvements. This research may be suitable for publication in Electronics after that. Here are some points to consider:

1) The number of conference references is very high. I suggest to include more journal or book references. Additional comments related to lateral control issues are needed. For instance, there is a significant concern regarding the necessity to observe vehicle states that cannot be directly measured, such as roll and sideslip angles. These states are vital when designing steering controllers. Furthermore, given the numerous disturbances that impact vehicle dynamics, it is imperative to ensure that the closed-loop system is stable and robust against these effects. The authors are advised to address these issues in the introduction. As a suggestion, you can review the following references and discuss their relevance if deemed appropriate: https://doi.org/10.3390/machines12010053, https://doi.org/10.1109/TITS.2023.3321415 

2) From Figure 14, I see that the stationary lateral displacement error is never zero for the proposed controller. Moreover, this stationary error varies from one driver to another. In my experience, achieving zero steady-state lateral error for any path-following problem is always aided by the integral gain of the PID. This is a rare occurrence. Please provide further details on it.

3) Please provide further details on the ITAE index and how it is calculated. Please include its full name in the paper (integral of time-multiplied absolute value of error). In addition, it is important to include the units for each value obtained from the ITAE, as the ITAE does no return non-dimensional numbers.

4) For the test depicted in Figure 18, the vehicle speed is 53.8 m/s, which equals to 193.68 km/h. Why did you choose a tough test rather than a more realistic situation such as 120 km/h? When simulating vehicle behaviour at high speeds, more complex vehicle models are required to provide reliable results, at the cost of additional computational resources. Please add further details on the simulation configuration in order to carry this tests in real time. (Vehicle dynamics model, Simulation time step, vehicle model update frequency, controller frequency…)

5) Please include some relevant numerical indicators regarding the performance of the proposed system in the conclusion section. 

Minor concerns

1) This text requires a review of its use of the English language.

2) Figure 1 quality must be enhanced. There are some incomplete textboxes.

3) Figure legends should be contained within the figure (see Figures 14, 17 or 22 as an example).

Comments on the Quality of English Language

Editing of English language required

Author Response

RE: RESPONSE TO REVIEWER 1 COMMENTS

The authors sincerely appreciate the reviewer's invaluable comments regarding this review paper. This document shows how and where the comments have been addressed in the revised paper.

Point1: The number of conference references is very high. I suggest to include more journal or book references. Additional comments related to lateral control issues are needed. For instance, there is a significant concern regarding the necessity to observe vehicle states that cannot be directly measured, such as roll and sideslip angles. These states are vital when designing steering controllers. Furthermore, given the numerous disturbances that impact vehicle dynamics, it is imperative to ensure that the closed-loop system is stable and robust against these effects. The authors are advised to address these issues in the introduction. As a suggestion, you can review the following references and discuss their relevance if deemed appropriate: https://doi.org/10.3390/machines12010053, https://doi.org/10.1109/TITS.2023.3321415 

Response 1: Thank you very much for these constructive comments.

  • The conference reference number has been replaced with the journal papers. ([16], [20], [22], [33], [40-44], [46], [50], [51], [53], [65].

  • The roll and sideslip are factors that can lead to car accidents or fatal situations. These factors should be involved in modelling the car dynamic. We rely on the stability of the designed vehicle dynamics. Additionally, this study relies on designing a controller for various driving behaviours.

Point2:  From Figure 14, I see that the stationary lateral displacement error is never zero for the proposed controller. Moreover, this stationary error varies from one driver to another. In my experience, achieving zero steady-state lateral error for any path-following problem is always aided by the integral gain of the PID. This is a rare occurrence. Please provide further details on it.

Response 2: Thank you again for this valuable comment. The error varies or is never zero because the PID controller in this study does not form a closed loop on its own, and the driver movement considered as human in the loop is not always constant when driving. Additionally, the proposed controller aims not to remove the stationary error but to assist the driver in easily manoeuvring the steering wheel for diverse tasks.

Point 3: Please provide further details on the ITAE index and how it is calculated. Please include its full name in the paper (integral of time-multiplied absolute value of error). In addition, it is important to include the units for each value obtained from the ITAE, as the ITAE does no return non-dimensional numbers.

Response 3. Equation 9 in section 3.2.1 provided how the ITAE was calculated. The car designed in MATLAB provides the car X and Y coordinates, and the driving pathway data are available, so a function block in MATLAB has been designed to calculate the car's position distance to the centre line, and this distance is the position error. The orientation angle error is also calculated. After receiving those errors, the performance index integral time absolute error is determined, as shown in the figure below.  

The ITAE name and the units are mentioned in the text as requested.

Point 4:For the test depicted in Figure 18, the vehicle speed is 53.8 m/s, which equals to 193.68 km/h. Why did you choose a tough test rather than a more realistic situation such as 120 km/h? When simulating vehicle behaviour at high speeds, more complex vehicle models are required to provide reliable results, at the cost of additional computational resources. Please add further details on the simulation configuration in order to carry this tests in real time. (Vehicle dynamics model, Simulation time step, vehicle model update frequency, controller frequency…)

Response 4: Thank you for this comment. The speed units generated by the car dynamic are in miles/hour; it was my mistake, and the paper has been revised accordingly.

The simulation details were added in section 4.4

Point 5: Please include some relevant numerical indicators regarding the performance of the proposed system in the conclusion section. 

Response 5: The conclusion was revised, and some numerical indicator was added as requested.

Minor concerns

  • This text requires a review of its use of the English language.

Response : The English language was revised

  • Figure 1 quality must be enhanced. There are some incomplete textboxes.

Response: The figure has been improved as requested

  • Figure legends should be contained within the figure (see Figures 14, 17 or 22 as an example).

Response : The legend has been included in most of the figures. Some figures indicated The legend in the text below the figure because they blocked the curve.

Reviewer 2 Report

Comments and Suggestions for Authors The current version can be accepted.

Author Response

RE: RESPONSE TO REVIEWER 2 COMMENTS

The authors sincerely appreciate the reviewer's invaluable comments regarding this review paper.

The reviewer approved the submitted version of my paper and did not make any comments.

Round 2

Reviewer 1 Report

Comments and Suggestions for Authors

The manuscript has been improved. Congratulations for your work

Comments on the Quality of English Language

N/A